# A resource to empirically establish drug exposure records directly from untargeted metabolomics data

Despite extensive efforts, extracting medication exposure information from clinical records remains challenging. To complement this approach, here we show the Global Natural Product Social Molecular Networking (GNPS) Drug Library, a tandem mass spectrometry (MS/MS) based resource designed for drug screening with untargeted metabolomics. This resource integrates MS/MS references of drugs and their metabolites/analogs with standardized vocabularies on their exposure sources, pharmacologic classes, therapeutic indications, and mechanisms of action. It enables direct analysis of drug exposure and metabolism from untargeted metabolomics data, supporting flexible summarization at multiple ontology levels to align with different research goals. We demonstrate its application by stratifying participants in a human immunodeficiency virus (HIV) cohort based on detected drug exposures. We uncover drug-associated alterations in microbiota-derived *N*-acyl lipids that are not captured when stratifying by self-reported medication use. Overall, GNPS Drug Library provides a scalable resource for empirical drug screening in clinical, nutritional, environmental, and other research disciplines, facilitating insights into the ecological and health consequences of drug exposures. While not intended for immediate clinical decision-making, it supports data-driven exploration of drug exposures where traditional records are limited or unreliable.

Chemical exposures play critical roles in shaping human health, with drugs representing a major component of the exposome[1]. According to a recent survey by the Center for Disease Control and Prevention, nearly half (45.7%) of the U.S. population reported using at least one prescription drug in the past 30 days[2]. Drug concentrations in human blood often reach levels comparable to those of endogenous and dietary molecules[3], and can significantly influence both the metabolic states and the microbiome compositions[4–7]. To assess drug exposure, clinical research relies on medical records, self-reports, or medication tracking (prescription counts, pill counts) strategies[8]. However, these methods are often costly, incomplete, and prone to recall bias[8–10]. They frequently overlook over-the-counter medications and dietary supplements,

and often fail to account for patient adherence. Moreover, they often miss undocumented drug use, including medications purchased online[10,11], acquired across borders[12,13], or consumed through secondary use of leftover drugs. Some exposures go entirely unrealized – for example, drugs introduced into the food supply, such as natamycin, an antifungal used both to treat eye infections and as a dairy preservative. Variability in drug metabolism and clearance further complicates exposure assessment, as some drugs are rapidly eliminated while others persist for weeks to months[14,15]. Outside of clinical settings, such as epidemiological monitoring of pharmaceuticals in wastewater, medical records are not available. These limitations highlight the need for direct data-driven approaches to screen drug exposures.

✉ e-mail: pdorrestein@health.ucsd.edu

Untargeted metabolomics offers the opportunity to empirically assess drug exposures directly from biological and environmental samples. However, liquid chromatography-tandem mass spectrometry (LC-MS/MS) based annotations, which rely on reference MS/MS library matches, are often difficult to interpret. For instance, an annotation may return a complex IUPAC chemical name like "(2 R,3S,4 R,5 R,8 R,10 R,11 R,12S,13S,14 R)−11-[(2S,3 R,4S,6 R)−4-(dimethylamino)−3-hydroxy-6-methyloxan-2-yl]oxy-2-ethyl-3,4,10-trihydroxy-13-[(2 R,4 R,5S,6S)−5-hydroxy-4-methoxy-4,6-dimethyloxan-2-yl]oxy-3,5,6,8,10,12,14-heptamethyl-1-oxa-6-azacyclopentadecan-15-one". A text search in the right reference resource will hopefully link this IUPAC name to the common name, in this case "azithromycin". A second search is required to connect the name to its role as a drug, in the example case, an antibiotic originally isolated from a bacterium. While this example involves a simple name and a limited number of identifiers, other compounds, like penicillin G or aspirin, have hundreds of synonyms and identifiers in chemical databases such as PubChem, making the identification process more challenging. This task must be repeated for every obtained annotation, which can range from hundreds to thousands in a given untargeted metabolomics experiment, to find all drugs in a dataset.

Even after drug identities are resolved, interpreting their biological significance remains a challenge. Researchers must often conduct extensive literature and web searches to understand the therapeutic roles of the drugs and their mechanisms of action. Public databases, such as DrugBank[16,17], DrugCentral[18], DailyMed[19], and KEGG DRUG[20], can assist in interpretation. However, the pharmacologic information is often provided as plain text or combinatorial classifications that require manual organization before downstream analysis. Although large language models or similar text mining strategies can assist, the results still need manual verification to confirm accuracy[21–23]. In addition, it is common that only metabolized versions of a drug are present in the sample, leading to missed drug exposure if only the parent drug is considered[24]. Unfortunately, with very few exceptions, existing MS/MS reference libraries capture only the parent drug due to challenges in sourcing drug metabolite reference standards[25], leaving a gap in the current drug screening capacities.

The lack of MS/MS references for drug metabolites, combined with the complexity in interpreting annotation results, makes it very difficult to accurately and efficiently annotate all drug exposures in untargeted metabolomics studies. For instance, stratifying a cohort based on antibiotic exposure - perhaps to better understand microbiome changes - requires identifying all antibiotics and their metabolites present in the samples and understanding their mechanisms of action. This is currently challenging due to the lack of resources that provide objective, systematic, and efficient readouts of drugs in untargeted metabolomics experiments.

In this work, we curate the Global Natural Product Social Molecular Networking (GNPS) Drug Library to enable data science strategies on drug readouts. GNPS Drug Library contains a collection of MS/MS reference spectra for drugs and their metabolites/analogs along with standardized, machine-readable pharmacologic metadata for each reference spectrum, encompassing exposure source, pharmacologic class, therapeutic indication, and mechanism of action. This resource enables data science analysis to empirically - and retroactively - determine drug exposures using untargeted metabolomics, supporting research across disciplines.

## Results

### Reference spectra collection for drugs and metabolites

The creation of this library involved three key steps: (1) collecting MS/MS spectra of drugs and their metabolites from publicly available MS/MS reference libraries; (2) finding MS/MS spectra analogs of those drugs in publicly accessible untargeted metabolomics data - to enhance coverage of the metabolized versions of drugs; and (3) linking each reference MS/MS spectrum to controlled-vocabulary metadata - the key component of this resource that facilitates efficient data interpretation (Fig. 1a).

The reference MS/MS spectra of drugs and their known metabolites were collected from two of the largest open-access mass spectral libraries, namely the GNPS Spectral Library[26] and MSⁿLib[27]. To identify drug spectra from these two resources, we first performed metadata enrichment for all compounds in GNPS and MSⁿLib by structural and name search against PubChem (for synonyms)[28], DrugCentral[18], the Broad Institute Drug Repurposing Hub databases[29], ChEMBL (for pharmacologic information)[30], and DrugBank (for pharmacologic information and the Anatomical Therapeutic Chemical Classification code)[16,17,31]. Based on the enriched metadata regarding clinical phases, all MS/MS spectra of drugs and compounds in clinical trials were compiled into the centralized GNPS Drug reference library (see method details in Supplementary Text 1). This represents all publicly accessible drug MS/MS spectra, covering 4723 unique drugs (represented by 99,122 MS/MS spectra) analyzed with diverse instrumentation and collision energies. The compound names in the GNPS Drug Library were automatically curated and set to the first synonym in PubChem. We point out that the term "drug" is used here in a broad sense, as the GNPS Drug Library includes not only prescribed and over-the-counter medications but also compounds currently in clinical trials, drugs that have been withdrawn, as well as substances with potential for abuse (e.g., cocaine, fentanyl).

To ensure coverage of drug metabolites, we performed a second "partial name match" to include previously missed metabolites. For example, by searching for the name "venlafaxine" (an antidepressant), we obtained reference spectra for five of its metabolites, including "N-desmethylvenlafaxine", "O-desmethylvenlafaxine", "N,O-didesmethylvenlafaxine", "N,N-didesmethylvenlafaxine", and "venlafaxine N-oxide" that were not captured in the previous database matching. This approach allowed us to add reference spectra for metabolites of 110 drugs (2080 MS/MS spectra). Lastly, we also included the MS/MS spectra collected in the development of dmCCS[32], a collision cross-section database for drugs and their metabolites where human liver microsomes and S9 fractions were used for in vitro generation of drug metabolites. In total, we collected metabolite spectra for 470 drugs in the GNPS Drug Library (represented by 2080 MS/MS spectra; Fig. 1a).

### Drug analog propagation from public data

The extensive collection effort yielded drug metabolite reference spectra for 10% of all the drugs included in the GNPS Drug Library, leaving most drugs without spectra for their metabolites. To overcome this challenge, we hypothesized that unannotated drug metabolites are present in public untargeted metabolomics datasets. We further hypothesized that spectral alignment strategies can be used to find the modified versions of the drugs[33–35]. In other words, public untargeted metabolomics datasets could be used to create a reference library of candidate drug metabolites that will facilitate the drug exposure readout in future studies.

Based on MS/MS spectral alignment using two computational methods: repository-scale molecular networking[36] and fast Mass Spectrometry Search Tool (fastMASST) with analog search[37,38], we retrieved all MS/MS spectra analogous to drugs from all data in the MetaboLights, Metabolomics Workbench, and GNPS/MassIVE, three of the largest public repositories for metabolomics data (covering ~3500 LC-MS/MS datasets)[26,39]. These spectra represent drug-related molecules potentially derived from metabolism (host or microbiome), abiotic processes, or adducts of the drugs from MS measurements. We obtained analogous MS/MS spectra for 24.6% of the 103,209 reference spectra for drug and drug metabolites (>19 million drug-analog spectral pairs).

In testing of the propagated analog library, we identified the need for additional filters to enhance relevance of the propagated

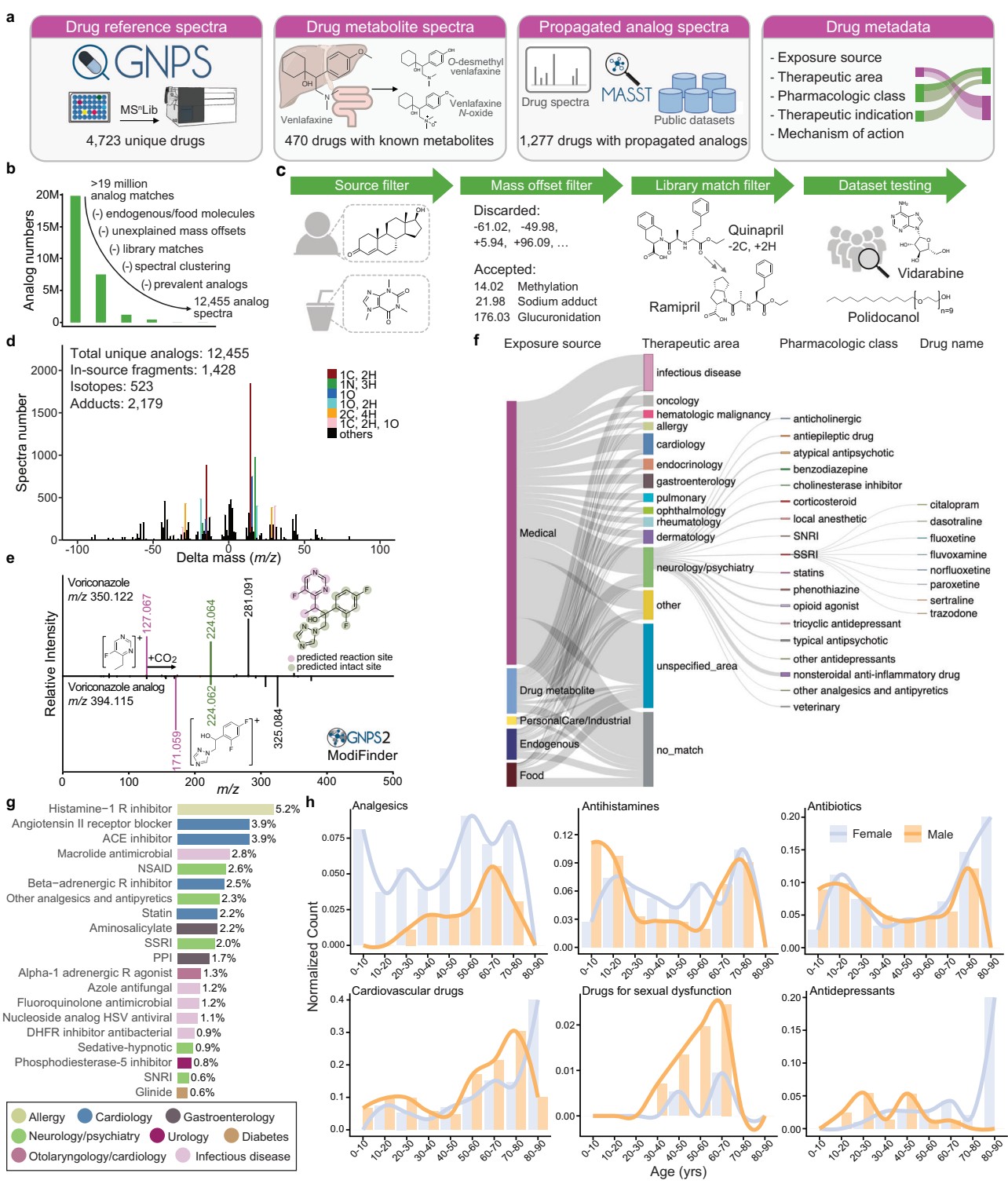

analog library to drug exposure (Fig. 1b, c). First, it is unlikely to determine the sources of exogenously supplied chemicals that can also be produced endogenously or derived from the diet. Consequently, structural analogs of drugs with endogenous or dietary sources were excluded from the propagated drug analog library (e.g., analogous MS/MS spectra of testosterone used to treat hypogonadism and for gender-affirming care, or caffeine used as a stimulant drug, were excluded). Second, propagated analogs with uncommon or unexplained mass offsets (precursor mass difference between the propagated analog and the paired drug) were excluded. Common mass offsets were collected from UNIMOD[40], from a

community-curated list of explainable delta masses (Supplementary Data 1), and from the Host Gut Microbiota Metabolism Xenobiotics Database[41], and were manually curated for those relevant to drug metabolism (e.g., 14.02 Da, methylation; 176.03 Da, glucuronidation) or mass spectrometry adducts (e.g., 17.03 Da, ammonium adduct; see Supplementary Data 2 for the 156 mass offsets that were included). Third, since drugs within the same pharmacologic family often have similar structures, they can be retrieved as analogs of each other through spectral alignments. Therefore, we excluded MS/MS spectra with matches to the whole public GNPS Library from the propagated analog annotations. For example, a

**Fig. 1 | The GNPS Drug Library and connected pharmacologic metadata. a** The GNPS Drug Library comprises four key resources: Drug MS/MS reference spectra, drug metabolite MS/MS reference spectra, propagated drug analogs derived from public metabolomics datasets, and pharmacologic metadata connected to each reference spectrum. **b** FastMASST analog search of drug spectra against public metabolomics studies yielded propagated drug-analogous MS/MS spectra, which were filtered by removing analogs for drugs with endogenous and food sources (source filter), removing mass offsets unexplained by common metabolic pathways (mass offset filter), removing analogs with GNPS library matches (library match filter), and removing analogs with unrealistic drug exposure indications (dataset testing). **c** Illustration of each filter employed in curating FastMASST analog match results. **d** Frequency of mass offsets in the propagated drug analog library. The mass offsets were grouped by unit mass and stacked based on the number of analog spectra. The most frequently observed mass offsets are colored while the rests are black. The numbers of in-source fragments, isotopes, and adducts in the drug analog library are estimated based on peak shape correlation and fragment

matching strategies (see Fig. S1 for details). **e** An example of structural modification sites predicted by ModiFinder[45]. Purple color highlights modified spectra and substructures, while the green color highlights unmodified ones. **f** Overview of the ontology-based drug metadata, highlighting common pharmaceutical classes and specific drugs in the neurology/psychiatry category. Width of the bars and lines reflects the number of unique drug structures in each class. **g** Barplot showing the top 20 most detected pharmacologic classes in fecal samples from the American Gut Project,[51] a cohort of the general population from the United States (US), Europe, and Australia (1993 individuals). Bars are colored based on the pharmacologic classes. **h** Detected therapeutic drug class patterns by age and sex (1845 individuals with age and sex information; age $46 \pm 18$ years [range 3–93], with 53% being female). Bars and lines are colored based on sex. NSAID non-steroidal anti-inflammatory drugs, ACE angiotensin converting enzyme, SSRI selective serotonin reuptake inhibitor, PPI proton pump inhibitor, HSV herpes simplex virus, DHFR dihydrofolate reductase, SNRI serotonin and norepinephrine reuptake inhibitor. Source data are provided as a Source Data file.

---

propagated analog of quinapril, an angiotensin converting enzyme (ACE) inhibitor, had a spectral match to ramipril, another ACE inhibitor (Fig. 1c). Excluding these analog annotations ensures that they do not overwrite library matches of known drugs and metabolites. Finally, we tested the propagated drug analog library against 20 public LC-MS/MS datasets to filter out analogs that have unrealistically high detection frequency. The selected datasets represent a broad range of human tissue types and biofluids, including fecal ($n = 5$), breast milk ($n = 2$), plasma ($n = 3$), skin ($n = 1$), and brain ($n = 1$), as well as multiple mouse tissues ($n = 8$). Here, we observed analogs of tocofersolan (a synthetic vitamin E derivative), iloprost (a synthetic prostacyclin mimetic), desonide (a synthetic topical corticosteroid), medroxyprogesterone (a synthetic progestin), and vidarabine (an adenosine analog used as an antiviral) in >50% of the human fecal samples from the American Gut Project ($n = 1993$ individuals), a cohort of the general population. The connected drugs for these analogs are derivatives of endogenous or food-derived molecules and are unlikely to be used by more than half of the population. Therefore, these analogs cannot be confidently linked to drug exposures and were excluded. Analogs of polidocanol (a synthetic long-chain fatty alcohol used as anesthetics) were observed in >70% of 2463 human milk samples. They are likely surfactants/contaminants with the polyethylene glycol structural units[42] and thus were excluded from the propagated drug analog library (Fig. 1c).

After the filtering steps, propagated analogs of 1277 drugs (12,455 clustered non-duplicated MS/MS spectra) were collected in the final drug analog library. We observed that 63% of the propagated analogs occurred at least once with the corresponding parent drugs in the same data file, highlighting the relevance of the propagated drug analogs to drug exposures (Fig. S1). The most common mass offsets in the drug analog library correspond to a gain or loss of 14.02 Da, which can be interpreted as (de)methylation, followed by a gain of 17.03 Da ($NH_3$, ammonium adduct), a gain of 15.99 Da (oxygen, oxidative metabolism), a loss of 18.01 Da ($H_2O$, dehydration), and a loss of 28.03 Da ($C_2H_4$, (de)ethylation; Fig. 1d). Based on peak shape correlation and fragment matching analysis[43,44], 33% of the propagated drug analogs could be other ion forms of the parent drug, including isotopes (5%), adducts (17%), or in-source fragments (11%), while the rest 67% are likely drug metabolites or structural analogs (Fig. 1d, S1). Although the propagated analogs include non-biological ion forms such as adducts and in-source fragments, we retain them in the library to enhance detection sensitivity. Their presence still signals drug exposure even if they do not represent metabolic derivatives. To extend structural hypotheses for the drug analogs that we found, we employed the newly developed ModiFinder[45], which leverages the shifted MS/MS

fragment peaks in the MS/MS alignment to predict the most likely location of the structural modifications (Fig. 1e, S2). The source (as in-source fragments, adducts, or structural analogs) and the availability of ModiFinder prediction for each drug analog were provided in the GNPS Drug Library metadata (see Data Availability section).

## Metadata integration for drug interpretation

Connecting drug detections to their therapeutic indications typically requires expert knowledge and/or extensive literature searches. The GNPS Drug Library addresses this challenge by providing controlled-vocabulary metadata together with the specific drug annotations. This allows users to annotate all drugs in an untargeted metabolomics dataset and directly obtain a table with exposure sources, pharmacologic classes, therapeutic indications, and mechanisms of action of the drugs, with their structures and names in a data science-ready format (Fig. 1f, S3). Particularly, the "exposure source" information categorizes the drugs in a combination of five classes, namely medical, endogenous, food, personal care, and industrial sources, which was developed based on the source categorizations from the Chemical Functional Ontology (ChemFOnt) database[46] and curated manually - by parsing of web pages and scientific literature - to increase compound coverage and improve accuracy and consistency. This categorization allows distinguishing endogenous or food-sourced molecules from those used only as medications. Examples include deoxycholic acid, an endogenous molecule also used for liver disease, and lactitol, a food sweetener also used as a laxative. Using the GNPS Drug Library metadata, such annotations can be separated from those molecules used exclusively as drugs, which have distinct exposure implications.

Through structural and name matches, we then extracted the pharmacologic classes of 900 drugs from the U.S. Food and Drug Administration (FDA) and the therapeutic areas, therapeutic indications, and mechanisms of action for 3894 drugs from the Broad Institute Drug Repurposing Hub[29]. However, we noticed substantial variability in the extracted information (e.g., inconsistent therapeutic areas assigned to drugs within the same pharmacological class; the sulfonamide antimicrobials sulfamethizole, sulfamethazine, and sulfacetamide were categorized as infectious disease, gastroenterology, and ophthalmology, respectively), or insufficient metadata for several drugs (e.g., common therapeutic indications missing). Therefore, this metadata was further manually curated by expert clinical pharmacologists to enhance and clean up the information retrieved from databases. This manual curation increased the metadata coverage to 4560 drugs. Drugs without associated metadata are typically those that have been withdrawn from the market (e.g., indoprofen), were in drug development but never marketed (e.g., tarafenacin), or are under

development but do not yet have regulatory approval (e.g., firsocostat).

In total, 735 drugs in the GNPS Drug Library (38,001 spectra) were identified with endogenous or dietary sources. The final metadata of the drug library covers 27 unique therapeutic areas, 571 pharmacological classes, 920 therapeutic indications, and 823 mechanisms of action (Fig. 1f, S3). Therapeutic areas of neurology/psychiatry, infectious disease, and cardiology have the highest number of included drugs (Fig. 1f) and reference spectra (Fig. S3). We note that these incidences reflect the availability of the reference spectra but not the prevalence of these drugs in the general population. Combining the exposure source and therapeutic area, we noticed that fewer drugs related to infection and neurology/psychiatry have endogenous or food sources, while higher portions of drugs used for gastroenterology (e.g., deoxycholic acid, riboflavin) and dermatology (e.g., salicylic acid, nicotinamide) are endogenous and/or food-derived molecules.

## Metabolites and propagated analogs enhance drug detection
The GNPS Drug Library can detect drugs known to be consumed with enhanced sensitivity, provided by the drug metabolites and propagated analogs. To demonstrate this, we analyzed two pharmacokinetic datasets where healthy individuals received specific probe drugs followed by time-series sampling[47–49]. In the first study, 10 participants received a single oral dose of diphenhydramine[47]. The drug was not detected in plasma and skin samples before administration, but was detected in all individuals post-administration over the course of 24 h (Fig. S4a, b). In plasma, detection frequencies peaked at 1–2 h (Fig. S4a, b), aligning with the reported time to maximum concentration (~2 hours) for diphenhydramine[48]. In skin, peak detection occurred at 10–12 h (Fig. S4a, b), reflecting the delayed deposition to skin compared to plasma for orally administered drugs. In the second study, 14 participants received a cocktail of oral probe drugs, namely caffeine, midazolam, and omeprazole[49]. The parent drugs were detected in plasma from 100% (caffeine), 46% (midazolam), and 100% (omeprazole) of participants within 8 hours post-administration, but were not detected in fecal samples (Fig. S4c, e). Inclusion of drug metabolites and propagated analogs improved detection rates: midazolam detection rose to 69% in plasma and 7.1% in feces. Similarly, for omeprazole, fecal detection increased from 0% to 21.4% when considering its metabolites and analogs (Fig. S4d, f). Notably, at the 8 h time point, 61.5% of participants exhibited detectable omeprazole only through its metabolites or analogs in plasma - highlighting the value of including derivative forms in drug exposure assessments. Together, these results underscore that drug detection is both biofluid- and time-dependent, and that metabolites and propagated analogs enhance detection sensitivity across sample types and time points. More broadly, the results emphasize the need to establish drug exposures empirically in the context of the analyzed samples, as clinical records do not account for drug distributions and rarely consider the time between drug intake and sample collection.

Connected with public untargeted metabolomics data, the GNPS Drug Library can reveal distinct drug exposure profiles among individuals in different disease, age, and sex groups. For different disease studies, we used the human disease ontology identifier (DOID) curated in ReDU, a controlled-vocabulary metadata for public metabolomics datasets[50]. Samples from individuals with inflammatory bowel disease, Kawasaki disease, and dental caries were characterized by high detection frequencies of antibiotics (Fig. S5a). Skin swabs from patients with psoriasis - who are at increased risk for fungal infections - were characterized by the presence of antifungal agents. Samples from people with human immunodeficiency virus (HIV) showed a high frequency of antivirals, while samples from individuals with Alzheimer's disease contained cardiology and neurology/psychiatry drugs, aligning with expected medication use in this aging population given the well-established link between Alzheimer's and cardiovascular disease (Fig. S5a).

To investigate drug exposures among different age and sex groups, we profiled 1993 fecal samples from the American Gut Project[51], with participants from the United States (US), Europe, and Australia, with age 46 ± 18 years (range 3–93; 53% female). A total of 89 different drugs were detected; the most frequently detected pharmacologic classes included histamine-1 receptor antagonist (allergy), angiotensin II-receptor blocker (cardiology), ACE inhibitor (cardiology), beta-adrenergic receptor inhibitor (cardiology), statin (lipid-lowering), non-steroidal anti-inflammatory drug (NSAID; analgesics), and selective serotonin reuptake inhibitor (SSRI; antidepressant), which matches the most commonly prescribed drug classes in these regions (Fig. 1g)[52–54]. There were more drugs per individual noted in the US cohort compared to the European and Australian cohorts (chi-square test; $\chi^2$ (8, $n$ = 1903) = 44, $p$ = 5.6 × 10$^{-7}$, Fig. S5b). When connected with age and sex information, the drug detection agrees with the expected usage patterns of different drug classes (Fig. 1h). For example, cardiovascular drugs were detected more frequently with increasing age, while analgesics, antihistamines, and antibiotics were detected across all ages[55,56]. We also observed that analgesics, such as NSAIDs and paracetamol, were more frequently detected in females (chi-square test; $\chi^2$ (1, $n$ = 1958) = 15.4, $p$ = 8.5 × 10$^{-05}$), consistent with the literature[57,58], and that drugs for sexual dysfunction were detected predominantly in males. Overall, empirical drug readout using untargeted metabolomics, facilitated by the GNPS Drug Library, demonstrated good specificity among individuals with different diseases, ages, and sexes. We note that the re-analysis of public metabolomics datasets is based on MS/MS matches, thus corresponding to level 2/3 confidence according to the 2007 Metabolomics Standards Initiative[59]. Due to practical challenges to access sample extracts for >20 studies discussed in the above analyses, we were not able to further validate the annotations with analytical standards.

## Microbial origins of propagated drug analogs
The GNPS Drug Library can allow the discovery of previously uncharacterized drug metabolites. To illustrate this, we analyzed fecal samples from the HIV Neurobehavioral Research Center (HNRC) cohort ($n$ = 322; age 55 ± 12 years), which included both people with HIV ($n$ = 222) and people without HIV ($n$ = 100). Among the 17,729 unique MS/MS spectra detected, 643 were annotated with the GNPS Drug Library. After removing drugs that could be from endogenous or food sources (because we cannot assess whether they were given as a medication) and grouping annotations of drugs, metabolites, or analogs, 175 unique drugs remained. Antiretroviral drugs (ARVs; drugs for the treatment of HIV), cardiovascular drugs, and drugs for anxiety and depression were the most frequently detected categories (Fig. 2a, S6a). Although antiretroviral therapy (ART) - the combination of multiple ARVs to treat HIV - has led to high rates of viral suppression, individuals living with HIV continue to experience disproportionately high rates of depression and cardiovascular disease[60–63]. This is reflected in the frequent detection of antidepressants and cardiovascular drugs in the analyzed samples.

Interestingly, 38% of the drugs were annotated together with their metabolites or analogs. The drug metabolites/analogs often co-occurred with the parent drugs, validating the relevance of drug metabolites/analog detections to exposures of the drugs (Fig. 2a). For example, darunavir (an ARV) was observed with 22 analogs (Fig. S6b). Peak shape analysis indicated that five of the darunavir analogs are adducts or in-source fragments (as judged by overlapping retention times)[64], while the others represent unknown metabolites or derivatives of this drug (Fig. S6c, d). For the analogs that are not in-source fragments, 78–100% (median 98%) of their occurrences were together with the darunavir parent drug (Fig. S6b). The detection of darunavir analogs without the parent drug likely reflects complete metabolism at

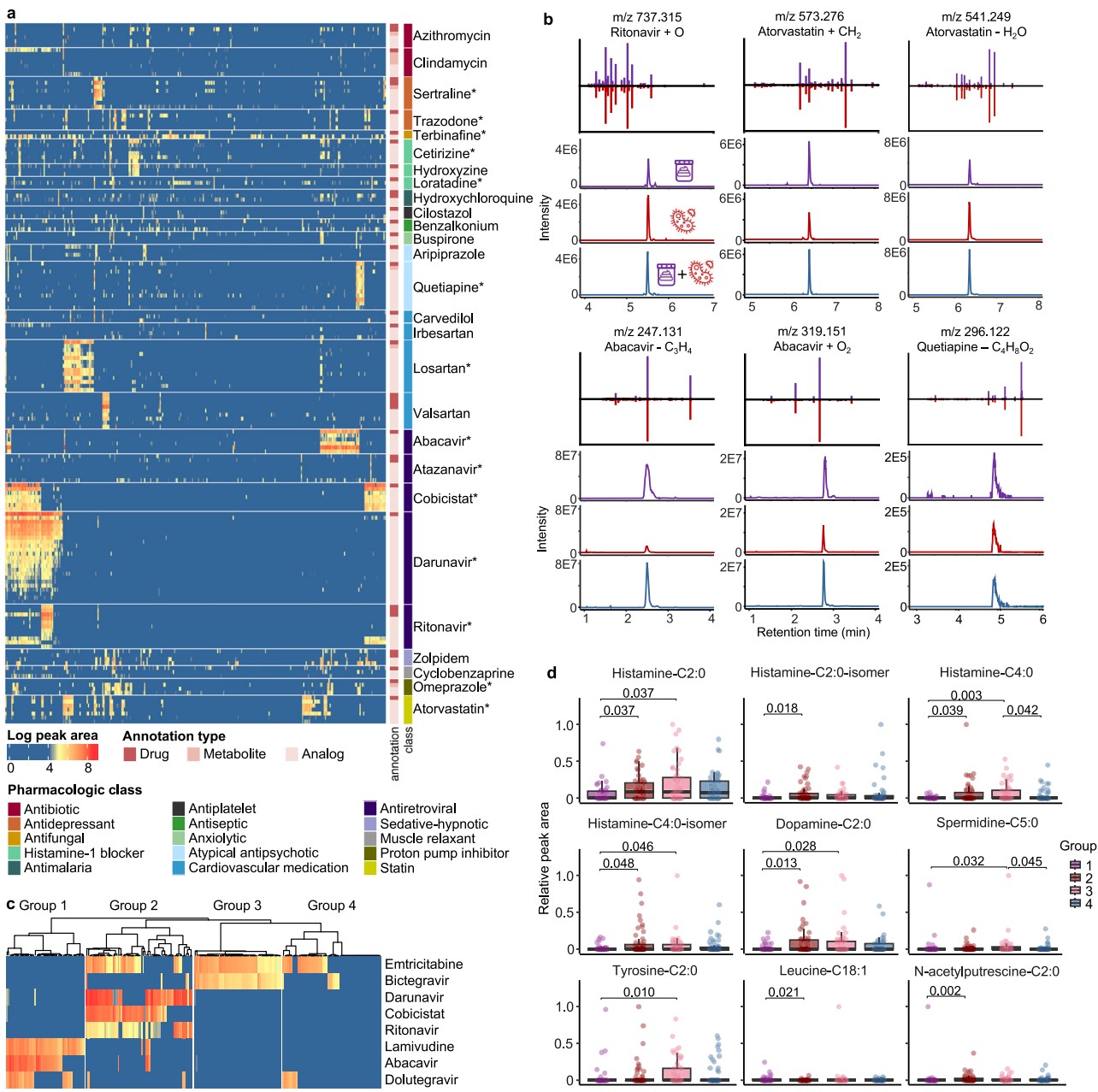

**Fig. 2 | Drug exposures in the HIV Neurobehavioral Research Center (HNRC) cohort with connections to microbial and endogenous metabolites.** From the HNRC cohort, 322 fecal samples were analyzed, with 222 samples from people with HIV and 100 samples from people without HIV. **a** Peak area visualization of drugs detected with at least two metabolites or analogs and in >10% samples. Each column represents one sample, and each row represents one drug annotation. Drug annotations were grouped based on the parent drugs and separated by gap spaces. Drug annotations were denoted based on their annotation types (as drug, drug metabolites, or drug analogs) and the pharmacologic classes of the parent drugs. All annotated ion/adduct forms of the parent drugs were visualized, leading to multiple rows of parent annotations for some drugs. Asterisks on the drug name mark parent drug annotations confirmed with commercial standards based on retention time and MS/MS spectral matches. Raw peak areas were log-transformed. **b** Retention time and MS/MS spectra mirror matches for drug analogs observed in both the fecal samples and the drug microbial incubations. Purple traces represent the fecal samples, while red traces represent the drug microbial incubation. Blue traces represent mixtures of the fecal samples and the microbial incubations at 1:1 volume ratio. The atomic changes of the drug analogs were based on [M + H]+ ion of the parent drug. **c** Hierarchical clustering of the samples from people with HIV (n = 222) based on detected antiretroviral drugs (ARV). Each row represents one detected ARV, with peak areas summed for the drug, metabolite, and analog detections followed by log-transformation (visualized with the same color scale as panel a). ARVs detected in <10% of samples are not shown. Each column represents one sample, clustered into four groups by hierarchical clustering with Ward's linkage and Euclidean distance. **d** Sample-to-sample peak areas of *N*-acyl lipids in people with HIV, separated by the clusters derived from ARV detections shown in (**c**) (Group 1, n = 47; Group 2, n = 64; Group 3, n = 52; Group 4, n = 59). For each compound, the peak area in each sample was standardized to the maximum value observed across all samples. A non-parametric Kruskal-Wallis test followed by pairwise Wilcoxon test and Benjamini-Hochberg correction for multiple comparisons was performed. *P*-values < 0.05 were noted in the figure. Boxplots showcase the median value, first (lower) and third (upper) quartiles, and whiskers indicate the error range as 1.5 times the interquartile range. Source data are provided as a Source Data file.

the time of sample collection, which, again, underscores the value of including drug metabolites and analogs in untargeted metabolomic screening to enhance the sensitivity and accuracy of drug exposure detection.

To further investigate the potential metabolic sources of the observed drug analogs, we cultured darunavir and 12 other drugs with a defined and complex synthetic microbial community of 111 bacterial species commonly found in the human gut[65]. To focus on drug analogs that are most relevant to this cohort, we included all drugs observed with three or more metabolites/analogs that were present in >10% samples (10 drugs in total). Omeprazole, loratadine, and terbinafine were additionally included because their analogs were frequently observed in samples without the respective parent drugs (Fig. 2a, Supplementary Data 3). Shared analogs were observed for 11 of the 13 drugs between the fecal samples and the microbial incubations. Among them, metabolites/transformation products were observed for 5 drugs (ritonavir, atorvastatin, abacavir, quetiapine, and omeprazole; Fig. 2b, S7), while the rest of the analogs were adducts or in-source fragments based on peak shape correlations[64]. The ritonavir, atorvastatin, abacavir, and quetiapine analogs increased in intensity with increased microbial incubation time (Fig. S7a–f), indicating gut microbial metabolism as a possible origin of the analogs and consistent with their observation in fecal samples. The omeprazole analog ($m/z$ 330.127) appeared to be an abiotic transformation product because it was already present at t = 0 cultures, and its intensity decreased with increased incubation time (Fig. S7g–i). This is consistent with the fast activation of omeprazole ($m/z$ 346.122), a proton-pump inhibitor and a prodrug, to the reactive sulphenamide product ($m/z$ 330.127) at low pH[66]. Rapid photolysis and hydrolysis of omeprazole have also been reported in abiotic environments with a major deoxygenation transformation product ($m/z$ 330.127)[67,68].

### Stratifying by drug exposure provides new biological insights

The GNPS Drug Library enables sample stratification based on drug profiles, facilitating the discovery of connections between drug exposures and metabolic changes. $N$-acyl lipids are a class of signaling molecules made by host-associated microbiota[69] that play important roles in the immune system[70], memory function[71], and insulin regulation of the human body[72–74]. Our recent work found that levels of histamine $N$-acyl lipids varied with HIV serostatus. Specifically, we observed elevated levels of histamine-C2:0, histamine-C3:0, and histamine-C6:0 in people living with HIV compared to those without HIV in the HNRC cohort[75]. To explore whether these differences were associated with drug exposures, we further stratified the samples based on their ARV exposure profiles. As anticipated, the ARV profiles aligned closely with HIV serostatus (Fig. S8a). High intensities of different ARVs were observed in fecal samples from people with HIV, while ARVs were only occasionally observed in people without HIV with low intensities. ARVs observed in people without HIV include tenofovir, atazanavir, and raltegravir, which are commonly prescribed for HIV prophylaxis (Fig. S8a)[76,77]. To control for HIV serostatus and investigate the effects of ARV exposure, we focused on samples from people with HIV ($n$ = 222) and further stratified them based on ARV co-occurrences. Hierarchical clustering revealed four distinct ARV exposure groups that corresponded well with common combination antiretroviral therapy (cART) regimens (Fig. 2c). For instance, Group 1 ($n$ = 47), characterized by lamivudine, abacavir, and dolutegravir exposures, corresponded to the dolutegravir/abacavir/lamivudine treatment regimen[78]. Group 2 ($n$ = 64) with emtricitabine, darunavir, ritonavir, and cobicistat exposures, agreed with the darunavir/ritonavir regimen[79] and the darunavir/cobicistat/emtricitabine/tenofovir regimen[80]. Group 3 ($n$ = 52), characterized by emtricitabine and bictegravir exposures, may be related to the bictegravir/emtricitabine/tenofovir treatment regimen[81]. Group 4 ($n$ = 59) exhibited either only emtricitabine exposure or no detectable ARV, potentially due to poor adherence, severe comorbidities, HIV elite control, or ARVs not included in the GNPS Drug Library or not amenable with LC-MS/MS detections (Fig. 2c).

Notably, histamine-C2:0 - previously associated with HIV serostatus[75] - along with eight other $N$-acyl lipids, showed significant differences across these four ARV exposure groups (Kruskal-Wallis test, $p$-value < 0.05; see specific $p$-value in Fig. 2d). This suggests that variation in ARV exposure, beyond HIV serostatus alone, may influence the levels of microbial-derived $N$-acyl lipids. We emphasize that these patterns would likely remain obscured without empirical drug read-outs from untargeted metabolomics. Clinical metadata may not document exposures to individual drugs and often do not provide quantitative information on the exposure levels. For example, meta-data for the HNRC cohort on current ARV usage, which is based on self-reports, documented drug usage as "ARV-naïve" (never received ARV), "no ARV" (no current ARV use), "non-HAART" (currently using less than three ARVs), and "HAART" (currently using three or more ARVs). Based on these classifications, no significant differences were observed for the $N$-acyl lipids detected in these samples (Fig. S8b). Without the empirical drug readout, enabled by the GNPS Drug Library, the effects of drugs on microbial $N$-acyl lipid levels would be overlooked.

## Discussion

We anticipate that the GNPS Drug Library will play a key role in clinical research studies by providing individualized, within-study relative quantitative drug exposure profiles. This capacity can enhance sample stratification and hypothesis generation, particularly for individuals with the same disease but differing medication regimens or pharmacokinetics, as demonstrated in the above analysis on ARV therapy. While the GNPS Drug Library is not intended for immediate clinical decision-making, this tool lays the groundwork for future integration of exposure data into precision medicine frameworks, such as studies of drug adherence, microbiome-drug interactions, and treatment heterogeneity. Despite the focus on human biospecimens in this manuscript, the GNPS Drug Library is broadly applicable for drug screening across diverse research disciplines, such as food and environmental monitoring, where medical records are not available. For example, GNPS Drug Library can be applied for wastewater-based epidemiology to monitor population-level health trends. In wastewater samples collected from March to June 2020, we observed seasonality of drug usage: the abundances of a cough suppressant (dextromethorphan) and antibiotics decreased over the months, while those of antihypertensives, antidepressants, HIV medications, and antiepileptics remained stable (Fig. S9a). GNPS Drug Library can also be applied in the context of food monitoring. Using untargeted metabolomics files from ~3500 foods/beverages collected in the Global FoodOmics Project[82], we observed antibiotics (e.g., ampicillin, tetracycline) in fish, beef, and turkey, as well as antiparasitic drugs (e.g., spinosad, thiabendazole) that are also used as fungicides or insecticides in fruit and vegetables (Fig. S9b–e). The mass spectrometry community will play a key role in the evolution of this resource through the continued deposition of reference libraries and expansion of the public metabolomics datasets for analog searches. By harnessing the power of public data and machine-readable metadata, we can unlock opportunities to deepen our understanding of the intricate relationships between xenobiotic exposure and human biological systems.

It is important to understand that the use of the GNPS Drug Library holds certain limitations. As with all MS/MS library matches in untargeted metabolomics, annotations from the GNPS Drug Library should be considered structural hypotheses derived from MS/MS spectral matches. To obtain definitive identifications, key drugs and metabolites of interest must be validated by authentic analytical standards through retention time and MS/MS comparison to achieve Level 1 identification[59], or by orthogonal strategies such as in vitro

culturing when authentic standards are not available, particularly for drug metabolites. Additionally, dosage of the exposed drugs should be derived with analytical standards, should the scientific question warrant this, as untargeted metabolomics profiling itself can only provide relative peak area comparison. The GNPS Drug Library can only capture drugs that are detectable in the specific biological matrix of choice (e.g., brain and urine samples will likely have different drug exposure readouts) and drugs that are ionizable with the chosen mass spectrometry setup. When constructing the drug analog library, we designed the filters to retain analog spectra that can be as confidently linked to drug exposure as possible, at the likely cost of excluding true positives. For example, metabolism pathways uncataloged in our customized mass offset list - such as those specific to substructures - will be excluded. As this is an evolving resource, we encourage the community to not only add to, but also report any inconsistencies in the library and the metadata they may notice.

Finally, it is important to note that we do not anticipate the GNPS Drug Library to play an immediate role in clinical decision-making based on untargeted metabolomics profiles. However, the concepts introduced in this study will advance the use of untargeted metabolomics for personalized exposure assessment, pharmacokinetic profiling, and linking of exposures to biological responses, serving as foundational steps toward the broader goal of precision medicine, particularly by enabling exposure-informed analyses of clinical phenotypes and cohort substructure. We envision the GNPS Drug Library as a valuable resource supporting a wide range of scientific fields, ultimately driving new insights into both the ecological and health-related impacts of drug exposures.

## Methods
### Ethics oversight
All datasets included in the manuscript were approved by the Institutional Review Board for Human Research, either by the University of California San Diego or University of Colorado, Boulder, and performed in accordance with the Declaration of Helsinki: American Gut Project, protocol no. 141853 and 12-0582; diphenhydramine study, protocol no. 191026; Cooperstown cocktail study, protocol no. 161940; HNRC, 172092. All subjects provided written informed consent.

### Reference spectra collection and curation
Reference spectra of drugs were collected from the GNPS spectral library[26] and the MS$^n$ library (accessed 03/2023) using a Python script developed for the MS$^n$Lib workflow[27]. Detailed steps are summarized in the Supplementary Text 1. Briefly, the chemical structures provided in the open-source libraries are first cleaned by removing salt forms and standardized with the ChEMBL structure pipeline Python package[83]. Based on this cleaned structure, other structural information is calculated, including the canonical and, if available, isomeric SMILES, InChI, and InChIKey strings. Database identifiers, e.g., DrugBank ID or ChEMBL ID, are searched in PubChem and Unichem based on the InChIKey strings. Those identifiers, the complete InChIKey or the first part of the InChIKey to remove stereochemistry, are then used to search in drug databases, including the Broad Institute Drug Repurposing Hub (downloaded July 2022)[29], DrugBank (downloaded August 2022)[16,17], DrugCentral (downloaded October 2022)[18], and the database of bioactive molecules from ChEMBL with clinical phases 1-4 (downloaded March 2023)[30]. Reference spectra with chemical structures contained in any of the databases above are retained in the GNPS Drug Library.

To increase the coverage for drug metabolites and derivatives, we additionally searched for reference spectra whose names contained the full name of a drug mentioned in the abovementioned databases, followed by manual inspection to remove mismatches. For example, by searching for the name "venlafaxine" (a serotonin and norepinephrine reuptake inhibitor indicated for the treatment of depression), we collected reference spectra for five metabolites of venlafaxine, including "N-desmethylvenlafaxine", "O-desmethylvenlafaxine", "N,O-didesmethylvenlafaxine", "N,N-didesmethylvenlafaxine", and "venlafaxine N-oxide" from the GNPS Library.

### Drug analog search and result filtering
Drug analog spectra were retrieved from the GNPS/MassIVE, MetaboLights, and Metabolomics Workbench public repositories based on MS/MS spectra alignment using two computational strategies: repository-scale molecular networking[36] and fastMASST with analog search[37,38]. A repository-scale molecular network was constructed (based on cosine similarity scores > 0.8 and a minimum of 6 matching ions)[36] and annotated with the 103,209 reference spectra of drugs and drug metabolites collected in this study. Unannotated spectra directly linked to annotated drugs or drug metabolites were collected as tentative drug analogs. FastMASST (fast Mass Spectrometry Search Tool)[37,38] checked the similarity of the queried spectrum against all spectra in the public repositories without pre-constructing a molecular network. Batch mode FastMASST search for the 103,209 reference spectra of drugs and drug metabolites was achieved in March 2025 with the customized Python scripts (https://github.com/robinschmid/microbe_masst), leveraging the Fast Search tool API[84]. The Python scripts returned analog matches as metabolomics universal spectrum identifiers (USIs)[85], connecting to spectra in the public repositories. Analog spectrum matches were accepted with a modified cosine score of 0.8 or higher and a minimum of 6 matching ions, with a search window of 200 Da below and above the precursor mass.

Filtering of the analog matches was achieved using customized R scripts with the following steps. *Step One*, analog matches to drugs with endogenous or dietary sources, based on the manually curated metadata, were excluded to avoid matches to endogenous metabolites. *Step Two*, analogs with mass offsets (precursor mass differences between the analogs and the drugs) unrepresented by common metabolism pathways were excluded. Specifically, mass offsets sourced from UNIMOD[40], from a community-curated list of delta masses (Supplementary Data 1), and from the Host Gut Microbiota Metabolism Xenobiotics Databases[41] were manually curated to only keep those relevant to drug metabolism (e.g., 14.02 Da, methylation; 176.03, glucuronidation) or mass spectrometry adducts (e.g., 17.03, ammonium adduct; see Supplementary Data 2 for the 156 mass offsets included). *Step Three*, analogs with matches to the full GNPS Library were excluded to restrict the analog library to true unknowns. Spectra of the remaining USIs from Step Two were downloaded using the metabolomics-usi API[85], clustered by the Falcon spectrum clustering tool (with default settings except an eps score of 0.05)[86], and searched against the full GNPS Library with the library search workflow (job link: https://gnps2.org/status?task=dabe0537e415416c9b2efadd9280b945). *Step Four*, the remaining drug analog libraries were tested against 20 public datasets to remove edge cases where the analogs were detected with high frequency and cannot be confidently linked to drug exposure. The testing datasets include five datasets of human fecal samples (GNPS/MassIVE ID: MSV000080673, MSV000095418, MSV000094515, MSV000092833, MSV000082493), two for human breast milk samples (MSV000091520, MSV000090877), three for human plasma (MSV000094395, MSV000085944, MSV000084008), one for human skin tissues (MSV000090877), one for human brain tissues (MSV000086415), and eight for mouse tissues (MSV000091363, MSV000096035, MSV000096036, MSV000096037, MSV000096038, MSV000096039, MSV000093230, MSV000093168). Search against GNPS Drug Analogs was performed using the Library Search workflow on GNPS with minimum cosine score of 0.7 and minimum 5 matching peaks, with job links provided in R scripts for the analog library development (see *Code availability* section).

To extend structural hypotheses for the drug analogs, we employed the newly developed ModiFinder[45], which leverages the shifted MS/MS fragment peaks in the MS/MS alignment to predict the most likely location for the structural modifications. ModiFinder was run with mass tolerance of 40 ppm and fragmentation depth of 2. ModiFinder predicted the partial location of modification for 32% of the analog spectra (60% with confidence score > 0.6). The job link is: https://gnps2.org/status?task=3f23c8ae79c34a6e86e7d64fc6919b76.

### Estimation of propagated analogs derived from analytical artifacts

To assess the proportion of propagated drug analogs derived from analytical artifacts versus those representing drug derivatives, we first classified the propagated analogs based on their mass differences with the parent drugs. Analogs with mass offsets of +1.00 or +2.00 Da were classified as isotopes, while those with offsets of +21.98, +37.95, and +37.96 Da were categorized as sodium, calcium, and potassium adducts, respectively. For the remaining analogs, we used peak shape correlations to identify propagated analogs sourced from analytical artifacts. We applied the fastMASST pipeline[37,38] to identify public mass spectrometry files that contain both the analog and its associated parent drug. MGF spectra files of the GNPS Drug Library were searched against the GNPS/MassIVE, MetaboLights, Metabolomics Workbench repositories using the following parameters: precursor and fragment $m/z$ tolerances of 0.02 Da, cosine similarity threshold of 0.9, and a minimum of 6 matched peaks. For each detected analog-drug pair, extracted ion chromatograms (XICs) of the drugs and analogs were constructed as ten consecutive MS1 scans centered around the peak apexes. Pearson correlation analysis was then performed to assess alignment between the XIC intensity profiles. Based on the maximum Pearson correlation coefficient ($R^2$) observed across multiple files, the propagated analogs were classified as analytical artifacts ($R^2 > 0.9$) or drug derivatives ($R^2 < 0.9$). Finally, analogs that did not co-occur with the parent drugs were classified based on MS/MS fragment evidence. Specifically, analogs with higher precursor $m/z$ than the parent drug were identified as adducts if any of their fragment ions matched the drug's precursor $m/z$. Conversely, analogs with lower precursor $m/z$ were assigned as in-source fragments if their precursor $m/z$ matched any of the drug's fragment ions. No intensity filtering was applied to the MS/MS spectra in this classification step.

### Metadata curation

The GNPS Drug Library provides ontology-based metadata on the exposure sources, therapeutic areas, pharmacologic classes, therapeutic indications, and mechanisms of action of the drugs connected to each reference spectrum. The exposure source information was first retrieved from the ChemFOnt database[46] through name matching. Synonyms of the drugs in the GNPS Drug Library were extracted from PubChem with the MS$^n$Lib Python scripts and searched against chemical names in the ChemFOnt database. The results were then manually curated by three experts by parsing web pages and scientific literature to increase coverage to 100% of the spectra in the GNPS Drug Library and improve accuracy and consistency. The therapeutic areas, mechanisms of action, and therapeutic indications of the drugs were extracted from the Broad Institute Drug Repurposing Hub[29] using the MS$^n$Lib Python scripts through InChIKey matches. The pharmacologic classes were extracted from the U.S. Food and Drug Administration (FDA) documents by synonym matching. The above database search returned the therapeutic areas, mechanisms of action, and therapeutic indications of 3,894 drugs (86,364 spectra) and the pharmacologic classes of 900 drugs (22,935 spectra). The extracted metadata were then manually curated by an expert pharmacologist to improve accuracy and fill in missing information to the extent possible, which increased the metadata coverage to 4,560 drugs (90,325 spectra). Sources used to curate the metadata included the Anatomical

Therapeutic Chemical (ATC) Classification, National Institutes of Health (NIH), FDA, European Medicines of Agency (EMA), and Drug-Bank. In particular, the WHO ATC codes were used in the manual curation of the pharmacologic class. The refined metadata can be illustrated with doxycycline, a tetracycline-class drug. Initially, it was classified under dental as a therapeutic area with periodontitis as the only indication. Following manual curation, doxycycline was reclassified under infectious diseases (therapeutic area) and Gram-positive and Gram-negative infections (therapeutic indication), recognizing its usage to treat a variety of infections. For drug metabolites and analogs, metadata corresponding to the parent drugs were used.

### Reanalysis of data from pharmacokinetic studies with known drug administration

In order to validate the empirical drug records from the GNPS Drug Library, we re-analyzed public datasets (MSV000085944, MSV000084008, and MSV000082493) from two healthy cohorts receiving specific drugs followed by intensive time-series sampling of multiple biofluids. The first cohort ($n = 10$) received a single dose of oral diphenhydramine (50 mg) with collection of plasma samples and skin swabs prior to (0 hour) and 24 hours after administration (0.5, 1, 1.5, 2, 4, 6, 8, 10, 12, and 24 hours)[47]. The second cohort ($n = 14$) received a cocktail of orally administered probe drugs including caffeine (2 mg/kg), midazolam (0.075 mg/kg), and omeprazole (40 mg). Plasma samples were collected prior to drug administration and 5 min, 30 min, 1, 2, 4, 5, 6, and 8 h after the drug administration, while fecal samples were collected at day 0-9 of drug administration[49]. The peak list files (.mzML) were searched against the GNPS Drug Library using the library search workflow on GNPS2, with precursor and fragment ion mass tolerances of 0.02 Da, a cosine score threshold of 0.7, and minimum 2 matched peaks. Annotations with cosine score lower than 0.9 and matched peaks fewer than 5 were removed in the downstream analysis. The GNPS2 jobs are available at: https://gnps2.org/status?task=db0131ce1f0a43e0b22499061abceed3, https://gnps2.org/status?task=e8561a1aeb84485993aa74259bb7c3d4, https://gnps2.org/status?task=64fbaf7d3efc433fae69db0fcb5575d1.

### Reanalysis of public data in ReDU with different disease groups

In order to have an overview of the drug exposures among different disease groups, matches of the GNPS Drug Library with public datasets (see details in section *Estimation of propagated analogs derived from analytical artifacts*) were merged with ReDU ("Reanalysis of Data User Interface"), a metadata interface for public untargeted metabolomics data with curated controlled vocabularies[50]. This merged table was then filtered to contain only matches to human datasets ("9606|*Homo sapiens*" in the NCBITaxonomy column). After filtering, the unique combinations of disease status and body parts were extracted based on the "DOIDCommonName" and the "UBERONBodyPartName" columns. Detected drugs were connected to their therapeutic areas using the GNPS Drug Library metadata, separating the "infectious disease" drugs into antibiotics, antifungals, and antivirals. The heatmap was generated to visualize exposure profiles based on disease status by calculating the numbers of samples with drugs in a certain therapeutic area, normalized to the numbers of samples in each sample type with at least one drug detection. The heatmap was generated using the 'ComplexHeatmap' package in R (v 2.20.0) with the script provided in the code availability section.

### Reanalysis of the American gut project data

In order to overview drug exposure profiles among ages and genders, the mzML files from the public metabolomics dataset of the American Gut Project (MSV000080673) were re-processed with feature extraction in MZmine 3[87]. The data ($n = 2440$ fecal samples; $n = 67$ laboratory blanks) were originally acquired with LC-qTOF-MS/MS, and the following MZmine settings were used: For mass detection, the noise

levels were 1E3 for MS1 detection and 5E1 for MS2 detection. For chromatogram building, the mass tolerance was set as 0.0050 $m/z$ or 30 ppm, the minimum consecutive scans as 5, and the minimum height as 3E3. For local minimum search for chromatographic deconvolution, the minimum search range was 0.1 min, minimum ratio of peak top to side was 2, and maximum peak duration was 2.0 min. The peaks were de-isotoped within 3 ppm $m/z$ and 0.08 minutes retention time tolerances, then aligned with 0.0050 $m/z$ or 30 ppm mass tolerance and 0.3 minutes retention time tolerance. The feature list was filtered for minimum detections in 2 samples and exported as a feature quantification table (.csv) and an MGF spectra file. The feature quantification table was subsequently filtered with R scripts to remove features with average peak areas in samples lower than 3-fold of those in blanks. For the remaining features, peak areas in samples were considered detected only if they were higher than 3-fold of the average peak areas in blanks.

Annotations were performed with the GNPS Drug Library using the Feature-Based Molecular Networking (FBMN) workflow on GNPS2[88]. The spectra were filtered by removing MS/MS fragments within ±17 Da of the precursor $m/z$ and only keep the six most intense fragments in ±50 Da window. The spectra were searched against the GNPS Drug Library with precursor and fragment ion mass tolerances of 0.01 Da, a cosine score threshold of 0.7, and minimum 2 matched peaks. The GNPS2 job is available at: https://gnps2.org/status?task=ea8e959b9dd54549a5f39bf8a684fdd2. Annotations with cosine score lower than 0.9 and matched peaks fewer than 5 were removed in the downstream analysis. The remaining annotations were manually inspected and were accepted only if two or more major fragment ions were matched.

Demographic information of the samples was collected from ReDU controlled vocabulary metadata, including "AgeInYears" column for age, "BiologicalSex" column for sex, "Country" for country of residence, and "LatitudeandLongitude" for geographic coordinates. Samples from the intensive care unit (ICU) microbiome pilot (Qiita study 2136)[89] were excluded from the analysis because their drug exposure does not resemble the general population ($n = 93$). In cases where individuals had more than 1 sample ($n = 47$), only the first collected sample was included in the analysis. Only individuals from the US, Europe, and Australia were included in the analysis, due to the few samples from other regions ($n = 50$). Annotations were filtered based on the exposure source to exclude drugs with endogenous or diet-derived sources. Drug annotations were further grouped based on unique parent compound names, and the number of drugs detected per individual was counted to create the world map visualization (Fig. S5b). Geographic coordinates were available for $n = 1903$ individuals. The world map was generated using the "rworldmap" package in R with scripts provided in the code availability section. For the visualization based on pharmacologic classes (Fig. 1g, h), drug annotations were further grouped based on the pharmacologic class metadata provided in the GNPS Drug Library. Each pharmacologic class was counted only once per individual when multiple drugs under the same class were detected. Age groups were defined in 10-years intervals (nine intervals for age 0–90 years old), and individuals with no information on age and sex ($n = 148$) and individuals >90 years ($n = 2$) were excluded from the analysis. To adjust for the varying number of samples across age and sex, normalization was performed by dividing the number of observations by the total number of samples within each age group and sex.

### Extraction and LC-MS/MS profiling of human feces in the HNRC cohort
**Sample extraction.** The stool samples ($n = 322$) were prepared with a recently developed automatic pipeline for simultaneous metagenomics-metabolomics extractions[90]. Specifically for the metabolite extraction, swabs were added into Matrix Tubes (ThermoFisher

Scientific, MA, USA) containing 400 μL of 95% (v/v) ethanol and capped with the automated instrument Capit-All (ThermoFisher Scientific, MA, USA). The tubes were shaken for 2 min (1200 rpm, the SpexMiniG plate shaker) followed by centrifugation for 5 minutes (2700 g). The supernatant was then transferred to a deep well plate using an 8-channel pipette, concentrated to dryness with a vacuum centrifuge concentrator (room temperature; ~5 hours), and stored at −80 °C. Prior to instrumental analysis, the plates were redissolved (10 minute sonication) in 200 μL of 50% acetonitrile (v/v) with 100 μg/L sulfadimethoxine as the internal standard. Plates were centrifuged at 450 g for 10 minutes, and 150 μL supernatants were collected for instrumental analysis. Laboratory blanks ($n = 60$; as blank solvents without fecal materials) were included and processed following the same procedures.

**Instrumental analysis.** The fecal extracts were injected (5 μL) into a Vanquish ultra-high-performance liquid chromatography (UHPLC) system coupled to a Q Exactive quadrupole orbitrap mass spectrometer (Thermo Fisher Scientific, Waltham, MA). A Kinetex polar C18 column (150 × 2.1 mm², 2.6 μm particle size, 100 A pore size; Phenomenex, Torrance) was employed with a SecurityGuard C18 column (2.1 mm ID) at 30 °C column temperature. The mobile phases (0.5 mL/min) were 0.1% formic acid in both water (A) and ACN (B) with the following gradient: 0–1 min 5% B, 1–7 min 5–99% B, 7–8 min 99% B, 8–8.5 min 99-5% B, 8–10 min 5% B. The mass spectrometer was operated in positive heated electrospray ionization with the following parameters: sheath gas flow, 53 AU; auxiliary gas flow, 14 AU; sweep gas flow, 3 AU; auxiliary gas temperature, 400 °C; spray voltage, 3.5 kV; inlet capillary temperature, 269 °C; S-lens level, 50 V. MS1 scan was performed at $m/z$ 100–1500 with the following parameters: resolution, 35,000 at $m/z$ 200; maximum ion injection time, 100 ms; automatic gain control (AGC) target, 5.0E4. Up to 5 MS/MS spectra per MS1 scan were recorded under the data-dependent mode with the following parameters: resolution, 17,500 at $m/z$ 200; maximum ion injection time, 100 ms; AGC target, 5.0E4; MS/MS precursor isolation window, $m/z$ 3; isolation offset, $m/z$ 0.5; normalized collision energy, a stepwise increase from 20 to 30 to 40%; minimum AGC for MS/MS spectrum, 5.0E3; apex trigger, 2 to 15 s; dynamic precursor exclusion, 10 s.

**Feature extraction and data analysis.** The raw spectra were converted to mzML files using MSconvert (ProteoWizard)[91] followed by feature extraction using MZmine 4[87] with the following parameters: For mass detection, the noise factor was 3.5 for MS1 and 2.5 for MS2. For chromatogram building, the mass tolerance was set as 0.01 $m/z$ or 10 ppm, the minimum consecutive scans as 3, and the minimum height as 5E2. Chromatograms were smoothed with the Savitzky Golay algorithm, followed by local minimum search for chromatographic deconvolution with minimum search range of 0.2 min, minimum ratio of peak top to side of 2, and maximum peak duration of 0.5 min. The peaks were de-isotoped within 5 ppm $m/z$ and 0.05 minutes retention time tolerances, aligned with 0.01 $m/z$ or 20 ppm mass tolerance and 0.3 minutes retention time tolerance, then gap-filled with 20 ppm $m/z$ tolerance and 0.3 min retention time tolerance. The feature list was blank-subtracted with 300% fold change increases compared with maximum blank values and 30 minimum number of detection in blanks. The feature list was then exported as a feature quantification table (.csv) and an MGF spectra file and used without post-filtering.

The features were annotated for drugs and drug analogs using the GNPS Drug Library with the same procedure as for the American Gut Project data. The GNPS2 job is available at: https://gnps2.org/status?task=d6f37a11d90c4f249974280c3fc90108. Peak areas of annotated drugs lower than 1E4 were replaced by zero to avoid false detections caused by instrument noises. Representative drug annotations, including all annotated ARVs and drugs used in the microbial incubations, were validated by analytical standards (Fig. S10). The features

were annotated for *N*-acyl lipids using a recently developed spectral library for 851 *N*-acyl lipids[75] with the following parameters: 0.02 Da precursor and fragment ion mass tolerance, a cosine score threshold of 0.7, and minimum 4 matched peaks (job link: https://gnps2.org/status?task=ee34ee95908749dd81ee9a62fbdac98e). The heatmaps to demonstrate drug annotations were generated using the 'Complex-Heatmap' package in R (v 2.20.0). People with HIV were clustered into four groups based on their ARVs profiles by hierarchical clustering under the Ward's linkage method and Euclidean distance matrix. Boxplots representing the peak areas of *N*-acyl lipids in different drug exposure groups were generated using the 'ggplot2' package in R (v 3.5.1), with R scripts provided in the *Code availability* section.

**Synthetic microbial community culture for drug metabolism Chemicals.** Drug standards for bacterial metabolism screening were purchased from Sigma-Aldrich (St. Louis, MO, USA; see detailed list in Supplementary Data 3). All solvents used were Optima LC-MS grade from Fisher Scientific (Pittsburgh, PA, USA). We note that quinine and quinidine were included in the drug metabolism assays because they were initially annotated based on MS/MS spectra matches (level 2/3 according to the 2007 Metabolomics Standards Initiative)[59]. After we acquired the standards, their retention times did not match the peaks in fecal samples. Therefore, quinine and quinidine were not considered in the downstream analysis.

**Synthetic microbial community culture.** The 111 bacterial strains used in this study are a selection of the human gut microbiota (hCom) previously published[65] that was optimized in the Zengler lab to accomplish uniform growth (Supplementary Data 4). A normalized working stock solution at $OD_{600} = 0.01$ was created for the 111 bacterial strains and stored at $-80\,°C$ until needed. For culturing, six groups were created and include the full hCom, the hCom-reduced (*Coprococcus comes* and *Coprococcus eutactus* were omitted due to their dominance in the community), fast growers, medium-fast growers, medium-slow growers, and slow growers (Supplementary Data 4). To create the hCom and hCom-reduced communities, the normalized stock of a bacteria was thawed, and a specific volume was taken and combined to generate the groups. To create the hCom-reduced, each bacteria was added based on their groups (fast-growers, 0.5 μL; medium-fast growers, 5 μL; medium-slow, 50 μL (not including the two species of *Coprococcus*); and slow growers, 500 μL). *Coprococcus comes* and *Coprococcus eutactus* were added at 0.5 μL to finalize the assembly of the full hCom. To create the other group communities, 500 μL of each bacteria of each group was combined (Supplementary Data 4). The groups were then diluted 1/100 at a final $OD_{600}$ of 0.0001 and cultured anaerobically in BHI medium (Supplementary Data 5) at 37 °C in an anaerobic chamber (10% $CO_2$, 7.5% $H_2$, 82.5% $N_2$).

**Drug metabolism assays.** A 10 mM stock solution was created for each drug and diluted to 200 μM with $H_2O$. Drugs were combined to create four different mixtures for the screening assay (mix1: sertraline, quinidine, venlafaxine; mix2: cetirizine, losartan, quetiapine, ritonavir; mix3: darunavir, quinine, atorvastatin, omeprazole; mix4: loratadine, terbinafine, trazodone, abacavir). Each drug was added to a final concentration of 2 μM (2 μL spike) after 24 h of bacterial acclimation. Cultures were immediately extracted after drug addition as the 0 h samples. Cultures from the fast and medium-fast groups were extracted after 48 h of growth, and the cultures from the remaining groups were stopped at 72 h. Medium controls were included by adding drug mixtures to the BHI medium without microbial communities, and were included at both 0 h and 72 h. Independent microbial incubations were performed in triplicate for each drug mix at both time points.

**Extraction of drugs and metabolites for LC-MS analysis.** Bacterial cultures were extracted using 600 μL of pre-chilled 50% (v/v) MeOH,

followed by an overnight incubation at 4 °C. Samples were dried using a CentriVap and stored at -80°C until LC-MS/MS analysis. Samples were resuspended in 200 μL of pre-chilled 50% (v/v) MeOH with 100 μg/L sulfadimethoxine as an internal standard, incubated at $-20\,°C$ overnight, centrifuged at 15,000 x g for 10 min, and 150 μL of the supernatant was transferred to 96 shallow well-plate for LC-MS analysis. Samples were analyzed with the same LC-MS/MS method for the HIV feces samples, except the following parameters: additional LC washing gradient after sample analysis, 8.5–9 min 5% B, 9–9.5 min 5–99% B, 9.5–10.5 min 99% B, 10.5–11 min 99–5% B, 11–12.5 min 5% B; MS1 scan range, *m/z* 100–100; AGC target, 1E6; dd-MS2 AGC target, 5.0E5; MS/MS precursor isolation window, *m/z* 1.

The raw spectra were converted to mzML files, followed by feature extraction using MZmine 4 with the following parameters: For mass detection, the noise factor was 3.0 for MS1 and 2.0 for MS2. For chromatogram building, the mass tolerance was set as 0.002 *m/z* or 10 ppm, the minimum consecutive scans as 4, and the minimum height as 1.5E5. Chromatograms were smoothed with the Savitzky Golay algorithm, followed by local minimum search for chromatographic deconvolution with minimum search range of 0.05 min, minimum ratio of peak top to side of 2, and maximum peak duration of 1.5 min. The peaks were de-isotoped within 3 ppm *m/z* and 0.04 minutes retention time tolerances, aligned with 0.002 *m/z* or 10 ppm mass tolerance and 0.07 minutes retention time tolerance, then gap-filled with 10 ppm *m/z* tolerance and 0.05 minutes retention time tolerance.

The features were annotated for drugs and drug analogs using the GNPS Drug Library with the same procedure as for the American Gut Project data. The GNPS2 job is available at: https://gnps2.org/status?task=401db749414546aeb37222c752294861.

**Co-migration analysis with fecal extracts.** Five samples from the bacterial cultures were analyzed again with selected fecal samples from the HIV infection cohort to confirm the retention time and MS/MS spectral matching of the drug analogs. One medium-slow grower at 72 h was used to confirm the ritonavir analog at *m/z* 737.31, one medium-fast grower at 72 h for the atorvastatin analogs at *m/z* 573.28 and *m/z* 541.25, one medium-fast grower at 72 h for abacavir analogs at *m/z* 247.13 and 319.15, one hCom culture at 72 h for the quetiapine analog at *m/z* 296.122, and one hCom-reduced culture at 0 hour for the omeprazole analog at *m/z* 330.13. Fecal samples with the highest analog intensities were used. Extracts of the bacterial cultures or feces were first diluted to similar concentrations with 50% (v/v) methanol based on peak areas of the drug analogs in the initial sample analysis, and then were mixed at 1:1 volume ratio. The selected bacterial culture samples, fecal samples, and mixture samples were co-analyzed using the same LC-MS/MS method for the bacterial cultures.

**Analysis of time-series samples of wastewater influents**
To demonstrate the utility of the GNPS Drug Library in public health settings, we analyzed wastewater influent samples (*n* = 55) collected from three wastewater treatment plants in Spain during the onset of the COVID-19 pandemic (March to June, 2020). The data were originally acquired with LC-Orbitrap-MS/MS and accessed from MSV000097575. Feature extraction was performed with MZmine 4[87] with the following parameters: For mass detection, the noise factor was 5 for MS1 and 2.5 for MS2. The chromatograms were built with mass tolerance of 0.002 *m/z* or 10 ppm, minimum consecutive scans of 4, and minimum height of 3E5, followed by local minimum search for chromatographic deconvolution with minimum search range of 0.1 min, minimum ratio of peak top to side of 1.8, and maximum peak duration of 1.5 min. The peaks were de-isotoped within 5 ppm *m/z* and 0.05 min retention time tolerances, aligned with 0.002 *m/z* or 10 ppm mass tolerance and 0.2 min retention time tolerance, then gap-filled with 5 ppm *m/z* tolerance and 0.1 min retention time tolerance. The features were then annotated for drugs and drug analogs using the

GNPS Drug Library, and the annotations were summarized using the GNPS Drug Library metadata. The GNPS2 job is available at: https://gnps2.org/status?task=4929dd75212743f48fa0383c5563456b.

## Statistical analysis

A chi-square test was performed to compare the number of drugs detected in fecal samples from different countries (Fig. S5b). Kruskal-Wallis tests followed by a pairwise Wilcoxon test were performed with Benjamini-Hochberg correction to compare the levels of *N*-acyl lipids in the four drug exposure groups (Fig. 2d, Fig. S8b) and the levels of drugs in wastewater influents from March to June (Fig. S9a). Adjusted *p*-values < 0.05 were considered as statistically significant. Pairwise Wilcoxon tests were performed to compare the drug analog intensities in bacterial cultures at 0 hour and 72 h (Fig. S7). Due to the small sample size ($n = 3$ in each group), *p*-values $\leq 0.1$ were considered as statistically significant.

## Reporting summary

Further information on research design is available in the Nature Portfolio Reporting Summary linked to this article.

## Data availability

The MGF spectral files for the GNPS Drug Library and the associated metadata of controlled vocabularies (.csv) have been deposited in Zenodo at https://doi.org/10.5281/zenodo.13892288[92]. The downloaded MGF spectral files can be added to personal GNPS folders and used directly for library matching. The raw data files for the pharmacokinetics studies have been deposited in GNPS/MassIVE repository (https://massive.ucsd.edu; same below) under the accession numbers MSV000085944, MSV000084008, and MSV000082493. Raw data files from the American Gut Project have been deposited at MSV000080673. Raw data files for fecal samples from the HNRC cohort have been deposited at MSV000092833. Raw data files for the drug bacterial cultures have been deposited at MSV000095331. Raw data files for HNRC fecal samples analyzed with the bacterial cultures have been deposited at MSV000096012. Raw data files for co-migration of the bacterial cultures and fecal samples have been deposited at MSV000096013. Raw data files for wastewater influents have been deposited at MSV000097575. Due to human subject protection constraints, clinical metadata for the HNRC cohort will be provided upon request to HNRC: https://hnrp.hivresearch.ucsd.edu. Source data are provided with this paper.

## Code availability

The code used to query reference spectra of drugs is available on GitHub under the MS$^n$ library project[27] (https://github.com/corinnabrungs/msn_tree_library; accessed 03/2023). The code used to query FastMASST api in batch mode to search for drug analogs is available on GitHub (https://github.com/robinschmid/microbe_masst; accessed 03/2025). The code used to filter the drug analog matches is provided on GitHub (https://github.com/ninahaoqizhao/Manuscript_GNPS_Drug_Library) and Zenodo (https://doi.org/10.5281/zenodo.17232336)[93]. The code used for dataset analysis can be found on GitHub (https://github.com/ninahaoqizhao/Manuscript_GNPS_Drug_Library and https://github.com/kinekvitne/manuscript_drug_library) and Zenodo (https://doi.org/10.5281/zenodo.17232336 and https://doi.org/10.5281/zenodo.17230320)[93,94]

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

## Acknowledgements

This project was enabled in part by the Alzheimer's Gut Microbiome Project (AGMP) and the Data Infrastructure and Molecular Atlas for AD: Connection Exposome, Gut Microbiome, and Metabolome supplement funded wholly or in part by the following grants thereto: 1U19AG063744 and 3U19AG063744-04S1 and awarded to Dr. Kaddurah-Daouk at Duke University in partnership with multiple academic institutions. As such, the investigators within the AGMP and the Exposome Supplement, not listed specifically in this publication's author list, provided data along with their pre-processing and prepared it for analysis, but did not participate in analysis or writing of this manuscript. A listing of AGMP Investigators can be found at https://alzheimergut.org/meet-the-team/. A complete listing of ADMC investigators can be found at: https://sites.duke.edu/adnimetab/team/. We also thank the support by NIH for the Maternal and Pediatric Precision in Therapeutics project P50HD106463, the development of tools for structure elucidation R01DK136117, and the Collaborative Microbial Metabolite Center U24DK133658. The HIV Neurobehavioral Research Center (HNRC) is supported by Center award P30MH062512 from NIMH. This research was supported in part by the Intramural Research Program of the NIH, National Institute of Environmental Health Sciences (ZIC ES103363). H.N.Z. was supported by the National Institute of Environmental Health Sciences of the National Institutes of Health under Award Number K99ES037746. C.B. was supported by the Czech Academy of Sciences PPLZ fellowship number L200552251. V.C.L. is supported by Fonds de recherche du Québec - Santé (FRQS) Postdoctoral fellowship (335368). N.E.A was supported in part by the National Center for Complementary and Integrative Health of the NIH under award number F32AT011475. A.M.C.-R. and P.C.D. were supported by the Gordon and Betty Moore Foundation grant GBMF12120. M.R. was supported by the NIH grant R37 AI126277. T.P. was supported by the Czech Science Foundation (GA CR) grant 21-11563 M and by the European Union's Horizon 2020 research and innovation programme under Marie Skłodowska-Curie grant agreement No. 891397. L.C., R.G.-S., and P.G.-F. were supported by the Spanish Ministry of Science (PID2022-139446OB-C21 and PID2022-139446OB-C22). L.C. acknowledges the support from the Economy and Knowledge Department of the Catalan Government through Consolidated Research Group (ICRA-TECH 2021 SGR 01283), as well as from the CERCA programme. W.B. acknowledges support by the Research Foundation–Flanders (FWO G0AHY25N).

## Author contributions

H.N.Z., C.B., and P.C.D. conceptualized the method. H.N.Z., C.B., R.S., and T.P. developed the MS/MS library for drugs. H.N.Z., W.B., R.T., and R.S. developed the MS/MS library for drug analogs. H.N.Z., S.Mohan, H.S., and P.R. curated exposure source metadata. K.E.K. curated pharmacological metadata. H.N.Z., R.T., Y.E.A., H.M.-R., N.E.A., A.M.C.-R., P.W.P.G., S.Magyari, I.M., A.C., S.P.T., S.Z. curated drug name match results. H.N.Z., K.E.K., S.Lamichhane, H.M.-R., L.K., and S.X. performed data analyses. Y.E.A., M.S.A., C.X.W., A.K.J., D.M., R.G.-S., P.G.-F., L.A.B. helped with data interpretation. V.C.-L., H.N.Z., L.Chin, C.W., M.R., K.Z. performed microbial incubation experiments. D.H.R., L.X. contributed MS/MS reference spectra. R.J.E., D.F., R.K.H., J.E.I., S.Letendre, and D.J.M. developed the clinical cohort of human immunodeficiency virus (HIV) infection. M.A., M.B., M.C.M., L.H., C.T., H.M.T., and J.Z. processed and acquired data on the fecal samples from the HIV cohort. M.R.Z.S., M.W. performed ModiFinder analysis. L.Corominas, R.G.-S., P.G.-F. collected wastewater samples. S.G., D.S.W., provided support on exposure

source annotation. R.K.-D. supervised the consortium providing access to the Alzheimer's disease cohort and acquired funding. R.K. supervised sample handling and DNA data acquisition for the Alzheimer's disease, HIV, and American Gut Project cohorts, and acquired funding. H.N.Z., K.E.K., C.B., and P.C.D. drafted the manuscript. P.C.D., S.M.T., acquired funding and supervised this project. All authors reviewed and edited the manuscript.

## Competing interests

R.S. is a co-founder of mzio GmbH. D.M.: D.M. is a consultant for BiomeSense, Inc., has equity, and receives income. The terms of these arrangements have been reviewed and approved by the University of California, San Diego in accordance with its conflict of interest policies. R.K.-D.: R.K.-D. is an inventor on a series of patents on use of metabolomics for the diagnosis and treatment of CNS diseases and holds equity in Metabolon Inc., Chymia LLC, and PsyProtix. M.W. is a co-founder of Ometa Labs LLC. T.P. is a co-founder of mzio GmbH. R.K. is a scientific advisory board member, and consultant for BiomeSense, Inc., has equity, and receives income. He is a scientific advisory board member and has equity in GenCirq. He has equity in and acts as a consultant for Cybele. He is a co-founder of Biota, Inc., and has equity. He is a cofounder of Micronoma and has equity and is a scientific advisory board member. He is a board member of Microbiota Vault, Inc. He is a board member of $N = 1$ IBS advisory board and receives income. He is a Senior Visiting Fellow of HKUST Jockey Club Institute for Advanced Study. The terms of these arrangements have been reviewed and approved by the University of California, San Diego, in accordance with its conflict of interest policies. S.M.T.: S.M.T. receives research funding from Veloxis Pharmaceuticals. P.C.D.: P.C.D. is a scientific advisor and holds equity in Cybele, and bileOmix, and is a Scientific Co-founder, advisor, holds equity and/or received income from Ometa, Arome, and Enveda, with prior approval by UC-San Diego. The remaining authors declare no competing interests.

## Additional information

Haoqi Nina Zhao [1,2,36], Kine Eide Kvitne [2,3,36], Corinna Brungs [4,5,36], Siddharth Mohan[2], Vincent Charron-Lamoureux [1,2], Wout Bittremieux [1,2,6], Runbang Tang[2], Robin Schmid [1,2,4], Santosh Lamichhane [2,7], Shipei Xing[1,2], Yasin El Abiead [1,2], Mohammadsobhan S. Andalibi [8,9,10], Helena Mannochio-Russo [1,2], Madison Ambre[11], Nicole E. Avalon [12], MacKenzie Bryant[11], Lindsey A. Burnett[13], Andrés Mauricio Caraballo-Rodríguez[1,2], Martin Casas Maya[11], Loryn Chin[14], Lluís Corominas [15], Ronald J. Ellis [8,9], Donald Franklin[9], Sagan Girod[16], Paulo Wender P. Gomes[1,2,17], Lauren Hansen[11], Robert K. Heaton[9], Jennifer E. Iudicello[9], Alan K. Jarmusch [1,2,18], Lora Khatib[8], Scott Letendre[10,19], Sarolt Magyari [2,20], Daniel McDonald[11], Ipsita Mohanty [1,2], Andrés Cumsille [2,21], David J. Moore[9,10], Prajit Rajkumar [2], Dylan H. Ross [22,35], Harshada Sapre[2], Mohammad Reza Zare Shahneh [23], Ruben Gil-Solsona[24], Sydney P. Thomas[1,2], Caitlin Tribelhorn[11], Helena M. Tubb[11], Corinn Walker[11], Crystal X. Wang [9,10], Jasmine Zemlin[1,2,25], Simone Zuffa [1,2], David S. Wishart [16,26], Pablo Gago-Ferrero[24], Rima Kaddurah-Daouk [27,28,29], Mingxun Wang[23], Manuela Raffatellu [11,25,30], Karsten Zengler [11,14,25,31], Tomáš Pluskal [4], Libin Xu [22], Rob Knight [11,25,32,33,34], Shirley M. Tsunoda [2] & Pieter C. Dorrestein [1,2,25] ✉

[1]Collaborative Mass Spectrometry Innovation Center, University of California San Diego, La Jolla, CA, USA. [2]Skaggs School of Pharmacy and Pharmaceutical Sciences, University of California San Diego, La Jolla, CA, USA. [3]Department of Pharmacy, University of Oslo, Oslo, Norway. [4]Institute of Organic Chemistry and Biochemistry of the Czech Academy of Sciences, Prague, Czechia. [5]Department of Pharmaceutical Sciences, University of Vienna, Vienna, Austria. [6]Department of Computer Science, University of Antwerp, Antwerp, Belgium. [7]Institute of Biomedicine and Turku Bioscience Centre, University of Turku and Åbo Akademi University, Tykistönkatu 6A, 20520 Turku, Finland. [8]Department of Neurosciences, University of California San Diego, La Jolla, CA, USA. [9]Department of Psychiatry, University of California San Diego, La Jolla, CA, USA. [10]HIV Neurobehavioral Research Program, University of California San Diego, La Jolla, CA, USA. [11]Department of Pediatrics, University of California San Diego, La Jolla, CA, USA. [12]Scripps Institution of Oceanography, University of California San Diego, La Jolla, CA, USA. [13]Department of Obstetrics, Gynecology and Reproductive Sciences, University of California San Diego, La Jolla, CA, USA. [14]Department of Bioengineering, University of California San Diego, La Jolla, California, USA. [15]Catalan Institute for Water Research (ICRA-CERCA), Girona, Spain. [16]Department of Biological Sciences, University of Alberta, Edmonton, AB, Canada. [17]Faculty of Chemistry, Federal University of Pará, Belém, PA, Brazil. [18]Immunity, Inflammation, and Disease Laboratory, Division of Intramural Research, National Institute of Environmental Health Sciences, National

Institutes of Health, Research Triangle Park, Bethesda, NC, USA. [19]Department of Medicine, University of California San Diego, La Jolla, CA, USA. [20]Institute of Microbiology, Eidgenössische Technische Hochschule (ETH) Zürich, Zürich, Switzerland. [21]Department of Plant Pathology, University of Wisconsin-Madison, Madison, WI, USA. [22]Department of Medicinal Chemistry, University of Washington, Seattle, WA, USA. [23]Department of Computer Science and Engineering, University of California Riverside, Riverside, CA, USA. [24]Department of Environmental Chemistry, Institute of Environmental Assessment and Water Research (IDAEA), Spanish Council of Scientific Research (CSIC), Barcelona, Spain. [25]Center for Microbiome Innovation, University of California San Diego, La Jolla, CA, USA. [26]Department of Computing Science, University of Alberta, Edmonton, AB, Canada. [27]Department of Psychiatry and Behavioral Sciences, Duke University, Durham, NC, USA. [28]Duke Institute of Brain Sciences, Duke University, Durham, NC, USA. [29]Department of Medicine, Duke University, Durham, NC, USA. [30]Chiba University, UC San Diego Center for Mucosal Immunology, Allergy and Vaccines (CU-UCSD cMAV), La Jolla, CA, USA. [31]Program in Materials Science and Engineering, University of California San Diego, La Jolla, CA, USA. [32]Department of Computer Science and Engineering, University of California San Diego, La Jolla, CA, USA. [33]Shu Chien-Gene Lay Department of Bioengineering, University of California San Diego, La Jolla, CA, USA. [34]Halıcıoğlu Data Science Institute, University of California San Diego, La Jolla, CA, USA. [35]Present address: Biological Sciences Division, Pacific Northwest National Laboratory, Richland, WA, USA. [36]These authors contributed equally: Haoqi Nina Zhao, Kine Eide Kvitne, Corinna Brungs. ✉e-mail: pdorrestein@health.ucsd.edu

