## [Transparent Peer Review file · Nature Communications]

A resource to empirically establish drug exposure records directly from untargeted metabolomics data

Corresponding Author: Professor Pieter Dorrestein

Version 0:

Reviewer comments:

Reviewer #1

(Remarks to the Author)

The authors did a great job to address my previous comments and now the paper have been largely improved. I have no further comments.

(Remarks on code availability)

Reviewer #2

(Remarks to the Author)

"A resource to empirically establish drug exposure records directly from untargeted metabolomics data"

This revised submission expands upon the earlier version, with additions such as an enlarged GNPS Drug Analog Library, an in-source fragment analysis, broader scope beyond clinical applications, and incorporation of plasma datasets alongside fecal metabolomics. While these changes address some prior concerns, they do not resolve several fundamental issues that limit the overall utility of the approach.

Validation of analog/metabolite identifications

The co-occurrence analysis (63% overall) is an indirect and insufficient validation strategy. Without systematic confirmation using authentic standards or orthogonal methods, there remains substantial risk of major misannotations, mis-assignment of in-source fragments, or detection of unrelated co-eluting compounds. The explanation that absence of the parent drug may reflect complete metabolism is speculative and unsubstantiated without direct evidence.

Applied and clinical relevance

While the authors now state the resource is "not for immediate clinical decision-making," the examples remain largely descriptive, with no compelling demonstration that the resource changes outcomes compared to existing approaches. The manuscript would benefit from a concrete, integrated example—such as showing how library-based exposure detection alters patient stratification, epidemiologic interpretation, or experimental conclusions. As it stands, the environmental and food monitoring examples feel tangential rather than woven into a coherent scientific narrative.

Sampling and detection gaps

The additional plasma analyses (Figure S5a) are retrospective and limited in scope, leaving unresolved the concern that fecal metabolomics misses certain exposures. Only a prospective, paired fecal/plasma study within the same cohort could convincingly address this limitation.

Quantification and dosage estimation

The authors acknowledge that untargeted metabolomics cannot provide dosage estimation, yet this absence substantially restricts the real-world applicability of the approach. No example is offered to show how this gap might be mitigated in practice, leaving a critical part of exposure interpretation unresolved.

Library expansion and quality control

The growth from ~3,200 to ~12,400 analog spectra is notable, but the manuscript lacks transparency on quality control: no clear confidence metrics, false discovery rates, or proportions of high-confidence vs. putative entries are provided. The continued inclusion of spectral counting, previously noted as largely meaningless, further detracts from the rigor. A breakdown of analog source types, validation levels, and associated confidence would be essential for users to gauge reliability.

Overall, while the authors have made additions, the work still suffers from fundamental weaknesses: indirect and incomplete validation, absence of quantification, limited evidence for practical impact, and insufficient transparency in library quality. These limitations (many acknowledged by the authors themselves) restrict the current utility of the approach for clinical, epidemiologic, or environmental applications.

(Remarks on code availability)
acceptable

**Point-by-point response to reviewer comments for:
A resource to empirically establish drug exposure records directly from untargeted metabolomics data**

Editor decision:

Your manuscript entitled "Empirically establishing drug exposure records directly from untargeted metabolomics data" has now been seen by 2 referees, whose comments are attached. I apologize for the delay in sending you this decision, due to difficulties in finding reviewers, and thank you for your patience. In the light of the reviewers' comments, I'm sorry to say that we have decided that we cannot offer to publish your manuscript in Nature Medicine.

While the referees find your work of some interest, they feel that the potential clinical application of the findings has not been sufficiently demonstrated, as well as raising substantial methodological concerns. Taken together, we feel that these reservations are sufficiently important as to preclude publication of this study in Nature Medicine.

However, I suggest that you consider Nature Communications as a suitable venue for this work. Please note that we have not discussed the manuscript with the editors at Nature Communications, but I would be happy to do so if you would like me to, in which case please let me know. To transfer a suitably revised version of the manuscript, please use our manuscript transfer portal. You will not have to re-supply manuscript metadata and files, unless you wish to make modifications. For more information, please see our manuscript transfer FAQ page.

Response: We really appreciate the effort from the editorial team and are very grateful for the chance to transfer our manuscript to *Nature Communications*. We have fully addressed the reviewer comments by (1) updating the GNPS Drug Analog Library by searching against the latest public metabolomics data, expanding the size of the analog library four-fold (from 3,234 to 12,455 clustered MS/MS spectra); (2) providing new analysis on the proportion of in-source fragments in the drug analog resources, with results shown in the new Figure S1; (3) largely revise the manuscript to discuss more on the clinical implication of the GNPS Drug Library, to emphasize that the resource is not intended for immediate clinical decision-making, and to showcase its utility beyond clinical research (see newly added Figure S9 for application in environmental epidemiology and food monitoring). We thank the reviewer for their thoughtful feedback, and we really appreciate this opportunity to improve our work. Please see below our detailed responses to the reviewer comments.

Reviewers' Comments:

Reviewer #1 (Remarks to the Author):

In this paper, the authors developed a Global Natural Product Social Molecular Networking (GNPS) Drug Library based on untargeted metabolomics using MS/MS spectra. The developed GNPS Drug Library comprises four key resources: Drug MS/MS reference spectra, drug metabolite MS/MS reference spectra, propagated drug analogs derived from public metabolomics datasets, and pharmacologic metadata connected to each reference spectrum. The authors also analyzed different datasets including pharmacokinetic datasets, public untargeted metabolomics data from patients with different disease conditions, and fecal metabolomics data from the American Gut project and a cohort of people with and without HIV to assess the utility of the GNPS Drug Library. In addition, through microbial incubation experiments, the potential metabolic sources of the observed drug analogs, i.e. gut microbial metabolism, were identified. Overall, this is an interesting study which provides very useful resources to objectively measure medication use and drug exposure using metabolomics data from human biospecimens, while clinical research relies on medical records and self-reported data. However, I have a few concerns about the accuracy of matched drug metabolites and analogs, fecal metabolomics data rather than blood metabolomics for drug exposure measures, and potential clinical implications.

Response: We thank the reviewer for supporting this study and the valuable inputs on our manuscript. We fully agree with reviewer concerns on the accuracy of the drug analogs, the use of fecal metabolomics data, and the lack of discussion on clinical implications. We have addressed these comments in the revised manuscript; please see detailed responses below. We note that, while dosage estimation remains beyond the scope of untargeted metabolomics, the empirical detection of drug exposures, particularly via co-occurrence with known parent compounds, enhances the fidelity of exposure profiling in clinical cohorts. This approach is especially valuable where chart data may be incomplete or unreliable.

Specific comments:

1. Given the limitations of untargeted metabolomics, the MS/MS spectra for drug metabolites may mis-annotate, and thus need further validation. In addition, it is unclear if the dosage of drug exposure can be indicated using untargeted metabolomics data.

Response: We fully agree with the reviewer that the MS/MS spectra for drug metabolites and analogs may mis-annotate. Therefore, we **validated the accuracy of the drug metabolites/analog**s based on their co-occurrence with the parent drugs, both across public datasets and in fecal samples from the HNRC cohort. Details were added in the revised manuscript.

Line 260-263: “We observed that 63% of the propagated analogs occurred at least once with the corresponding parent drugs in the same data file, highlighting the relevance of the propagated drug analogs to drug exposures (Figure S1).”

Line 387-398: “Interestingly, 38% of the drugs were annotated together with their metabolites or analogs. The drug metabolites/analog often co-occurred with the parent drugs, validating the relevance of drug metabolites/analog detections to exposures of the drugs (Figure 2a). For example, darunavir (an ARV) was observed with 22 analogs (Figure S6b) ... For the analogs that are not in-source fragments, 78-100% (median 98%) of their occurrences were together with the darunavir parent drug (Figure S6b). The detection of darunavir analogs without the parent drug likely reflects complete metabolism at the time of sample collection, which, again, underscores the value of including drug metabolites and analogs in untargeted metabolomics to enhance the sensitivity and accuracy of drug exposure detection.”

We also agree with the reviewer that **drug dosage** is critical information for clinical studies and untargeted metabolomics profiling is not able to provide this information directly. We now acknowledge this in the revised manuscript, and emphasize that the GNPS Drug Library is a tool to facilitate the screening/recognition of drug exposures. Once drugs are observed, then targeted assays should be developed to understand dosages of the drug exposures.

Line 514-516: “Additionally, dosage of the exposed drugs should be derived with analytical standards should the scientific question warrant this, as untargeted metabolomics profiling itself can only provide relative peak area comparison.”

2. Analyses using pharmacokinetic datasets showed that detection of drug metabolites using plasma samples was much better than using fecal samples. However, this study focused on fecal metabolomics to detect drug metabolites in human populations study, and thus many drug metabolites might be missed using fecal metabolomics. Blood metabolomics data from human cohort studies are needed.

Response: We fully agree with the reviewer that drug screening in blood is relevant information. In Figure S5a of the manuscript, we analyzed plasma samples from different disease types to investigate the drug exposure. For example, samples with Kawasaki disease and COVID-19 infection are characterized with antibiotics exposure (e.g., azithromycin, amoxicillin for Kawasaki disease; clarithromycin for COVID-19); samples with AID and HIV infection are high with antivirals (e.g., darunavir, efavirenz, ritonavir), while samples with Alzheimer’s disease are characterized with cardiology (e.g., atenolol, verapamil, valsartan) and neurology medications (e.g., donepezil, citalopram).

Beyond the provided blood analysis, we also note that from a clinical research standpoint, fecal samples offer a practical and mechanistically relevant matrix for investigating microbiome-drug interactions. Although plasma provides superior pharmacokinetic sensitivity, the inclusion of fecal data reflects real-world settings in which stool is more feasible to collect longitudinally or in decentralized cohorts. This makes it a

powerful adjunct in microbiome and exposome research. Inspired by this reviewer comments, we improved the analysis of the pharmacokinetic datasets by comparing the drug read-out with and without the drug metabolites/analogs. We noticed that the drug metabolites/analogs greatly enhanced the sensitivity of drug readouts in both plasma and fecal samples collected with a delayed period after drug administration. Therefore, we further emphasized the need to establish drug exposures empirically in the biofluids of interest with regards to specific research questions.

Line 325-339: “14 participants received a cocktail of oral probe drugs, namely caffeine, midazolam, and omeprazole. The parent drugs were detected in plasma from 100% (caffeine), 46% (midazolam), and 100% (omeprazole) of participants within 8 hours post-administration, but were not detected in fecal samples (Figure S4c,e). Inclusion of drug metabolites and propagated analogs improved detection rates: midazolam detection rose to 69% in plasma and 7.1% in feces. Similarly, for omeprazole, fecal detection increased from 0% to 21.4% when considering its metabolites and analogs (Figure S4d,f). Notably, at the 8-hour time point, 61.5% of participants exhibited detectable omeprazole only through its metabolites or analogs in plasma - highlighting the value of including derivative forms in drug exposure assessments. Together, these results underscore that drug detection is both biofluid- and time-dependent, and that metabolites and propagated analogs enhance detection sensitivity across sample types and time points. More broadly, the results emphasize the need to establish drug exposures empirically in the context of relevant biofluids – for example in feces when investigating microbiome-drug interactions - as clinical records do not account for drug distributions and rarely consider the time between drug intake and sample collection.

Revised Figure S4. Time-series drug detection in healthy individuals receiving specific drugs in pharmacokinetic studies. **a-b**, Detection frequencies of diphenhydramine in plasma and skin samples from 10 individuals receiving a single dose of oral diphenhydramine (50 mg), based on parent compound only (**a**) or parent compound, metabolites, and analogs (**b**). The line chart demonstrates detections before (0 hour) and 24 hours (0.5, 1, 1.5, 2, 4, 6, 8, 10, 12, and 24 hours) after drug administration. **c-d**, Detection frequencies of the administered drugs in plasma samples from 13 individuals receiving a cocktail of oral probe drugs, including caffeine (2 mg/kg), midazolam (0.075 mg/kg) and omeprazole (40 mg), based on parent compound only (**c**) or parent compound, metabolites, and analogs (**d**). Plasma samples were collected prior to (0 min) and 5 min, 30 min, 1, 2, 4, 5, 6, and 8 hours after administration of the drug cocktail. **e-f**, Detection frequencies of drugs in fecal samples from the 14 individuals receiving the drug cocktail, based on parent compound only (**e**) or parent compound, metabolites, and analogs (**f**). Caffeine was not detected in the fecal samples and therefore not included in panels e and f.

Lastly, to show the relevance of drug detection in more diverse sample matrices, we added an analysis on community-level health trends leveraging drug screening in wastewater.

We showed that, in wastewater influent samples collected during onset of COVID-19, the LC-MS/MS peak areas of cough suppressant medications and antibiotics decreased from March to June 2020, while the abundance of antihypertensives, antidepressants, HIV medications, and antiepileptics remain stable. The data is provided in **Figure S9a**.

Line 492-499: “In addition, the GNPS Drug Library is broadly applicable for drug screening across diverse research disciplines despite the focus on human biospecimen in this manuscript. For example, GNPS Drug Library can be applied for wastewater-based epidemiology to monitor population-level health trends. In wastewater samples collected from March to June 2020, we observed seasonality of drug usage - the abundances of a cough suppressant (dextromethorphan) and antibiotics decreased over time, while those of antihypertensives, antidepressants, HIV medications, and antiepileptics remain stable (Figure S9a).”

Revised Figure S9. Drug screening in wastewater and food samples using the GNPS Drug Library. a, Sample-to-sample peak areas of different drug classes in influent wastewater samples collected during March to June 2020 from three wastewater treatment plants (WWTPs) in Spain. Peak areas of multiple drugs in the same pharmacologic class were summed and standardized to the maximum value observed across all samples for this drug class. A non-parametric Kruskal-Wallis test followed by pairwise Wilcoxon test and Benjamini-Hochberg correction for multiple comparisons were performed. P-values < 0.05 were noted in the figure. Boxplots showcase the median value, first (lower) and third (upper) quartiles, and whiskers indicate the error range as 1.5 times the interquartile range.

3. It is expected that the GNPS Drug Library can reveal distinct drug exposure profiles across individuals with different disease, and metabolomics data are consistent with the expected drug usage of people with these diseases. However, data on individuals with the same disease but different drug usages might be more interesting and important for precision medicine research.

Response: We fully agree that stratification by drug exposure among individuals with the same diagnosis is a key advance toward personalized medicine. In our HIV cohort, we leveraged this capability to uncover treatment-related heterogeneity in microbiome-linked lipid signals—an example of how empirical drug profiling can elucidate mechanistic differences within clinically homogeneous populations. This is now further highlighted in the manuscript.

Line 486-489: “We anticipate that the GNPS Drug Library will play a key role in clinical research studies by providing empirical drug exposure profiles. This capacity can enhance sample stratification and hypothesis generation, particularly for individuals with the same disease but differing medication regimens, as demonstrated in the above analysis on ARV therapy.”

4. The rationale to perform gut microbial incubation experiment and to select darunavir and 12 other drugs in this experiment is not very clear. It might not be very relevant to the development of GNPS Drug Library and its utility described in this paper.

Response: We selected the 13 drugs for culturing based on high numbers of drug analogs observed and high prevalence of the drugs in this HNRC cohort. We agree with the reviewer that this experiment is not relevant to the development and utility of the GNPS Drug Library, but it **validates the drug analogs** and provides explanations for origins of some of the drug modifications. We further explained the rationale to select the 13 drugs in the revised manuscript.

Line 401-405: “To focus on drug analogs that are most relevant to this cohort, we included all drugs observed with three or more metabolites/analogues that were present in >10% samples (10 drugs in total). Omeprazole, loratadine, and terbinafine were additionally included because their analogs were frequently observed in samples without the respective parent drugs (Figure 2a, Table S3).”

5. The GNPS Drug Library provided very useful resources to study drug exposures using metabolomics data, but its potential clinical implications are not well-discussed. The potential role of this developed drug library in drug discovery and precision medicine is unclear, which needs further clarification.

Response: We fully agree with the reviewers that the clinical implications of the GNPS Drug Library are not well-discussed. A number of revisions were made accordingly.

First, we revise the sentence in the abstract, “*The GNPS Drug Library holds potential for broader applications in drug discovery and precision medicine*” to improve clarity.

Line 84-88: “Overall, GNPS Drug Library provides a scalable resource for empirical drug screening in clinical, nutritional, environmental, and other research disciplines, facilitating insights into the ecological and health consequences of drug exposures.

While not intended for immediate clinical decision-making, it supports data-driven exploration of drug exposures where traditional records are limited or unreliable.”

Second, we clarified that we anticipate the GNPS Drug Library to play key roles in clinical research studies, but likely not for immediate clinical decision-making based on untargeted metabolomics. We clarified the role of GNPS Drug Library as a metabolomics resource for precise drug exposure readout, and further discussed how this will facilitate the vision of precision medicine.

Line 486-492: “We anticipate that the GNPS Drug Library will play a key role in clinical research studies by providing empirical drug exposure profiles. This capacity can enhance sample stratification and hypothesis generation, particularly for individuals with the same disease but differing medication regimens, as demonstrated in the above analysis on ARV therapy. While it is not intended for immediate clinical decision-making, this tool lays the groundwork for future integration of exposure data into precision medicine frameworks, such as studies of drug adherence, microbiome-drug interactions, and treatment heterogeneity.”

Line 525-533: “Finally, it is important to note that we do not anticipate the GNPS Drug Library to play an immediate role in clinical decision-making based on untargeted metabolomics profiles. However, the concepts introduced in this study will advance the use of untargeted metabolomics for personalized exposure assessment, pharmacokinetic profiling, and linking of exposures to biological responses, serving as foundational steps toward the broader goal of precision medicine, particularly by enabling exposure-informed analyses of clinical phenotypes and cohort substructure. We envision the GNPS Drug Library as a valuable resource supporting a wide range of scientific fields, ultimately driving new insights into both the ecological and health-related impacts of drug exposures.”

Finally, we presented the broad implications of the GNPS Drug Library across diverse disciplines, including but not limited to clinical, nutritional, public health fields. We demonstrated this with two new analyses on drug screening in wastewater and diets.

Line 106-108: “Outside of clinical settings, such as epidemiological monitoring of pharmaceuticals in wastewater, medical records are not available. These limitations highlight the need for direct data-driven approaches to screen drug exposures.”

Line 492-503: “In addition, the GNPS Drug Library is broadly applicable for drug screening across diverse research disciplines despite the focus on human biospecimen in this manuscript. For example, GNPS Drug Library can be applied for wastewater-based epidemiology to monitor population-level health trends. In wastewater samples collected from March to June 2020, we observed seasonality of drug usage - the abundances of a cough suppressant (dextromethorphan) and antibiotics decreased over time, while those of antihypertensives, antidepressants, HIV medications, and antiepileptics remain stable (Figure S9a). GNPS Drug Library can also be applied in the context of food monitoring. Using untargeted metabolomics

files from ~3500 foods/beverages collected in the Global FoodOmics Project, we observed antibiotics (e.g., ampicillin, tetracycline) in fish, beef, and turkey, as well as antiparasitic drugs (e.g., spinosad, thiabendazole) that are also used as fungicides or insecticides in fruit and vegetables (Figure S9b-e).”

Revised Figure S9. Drug screening in wastewater and food samples using the GNPS Drug Library. **a**, Sample-to-sample peak areas of different drug classes in influent wastewater samples collected during March to June 2020 from three wastewater treatment plants (WWTPs) in Spain. Peak areas of multiple drugs in the same pharmacologic class were summed and standardized to the maximum value observed across all samples for this drug class. A non-parametric Kruskal-Wallis test followed by pairwise Wilcoxon test and Benjamini-Hochberg correction for multiple comparisons were performed. P-values < 0.05 were noted in the figure. Boxplots showcase the median value, first (lower) and third (upper) quartiles, and whiskers indicate the error range as 1.5 times the interquartile range. **b-e**, foodMASST search

outputs showcasing detections of drugs in food products. Pie charts display the proportion of MS/MS matches found in the deposited reference database. Blue indicates a match with a food sample, while yellow represents a non-match. GFOP, Global FoodOmics Project.

Reviewer #2 (Remarks to the Author):

This paper introduces the GNPS Drug Library, a resource designed to empirically assess drug exposure from untargeted metabolomics data. By compiling MS/MS reference spectra for drugs and metabolites alongside pharmacological metadata, the library aims to enhance drug use detection and complement traditional clinical records. While the concept has potential, significant shortcomings in its development and claims require critical examination. **Response:** We thank the reviewer for highlighting the potentials of our concepts and the valuable feedback on our manuscript. We have critically revised our manuscript following the reviewer comments. In particular, we provided detailed evaluation on the origins of the drug analogs as instrumental artifacts versus structural derivatives, showing that at maximum 33% of the drug analogs are derived from instrument artifacts, including but not limited to in-source fragments. We further emphasized the clinical and broad implications of this resource in the revised manuscript. Please see detailed responses below.

Overstated Claims - The authors position the GNPS Drug Library as transformative for precision medicine, yet its real-world clinical applicability remains entirely unvalidated. The claims of immediate utility are premature, as the tool has yet to demonstrate robustness in practical settings, such as across diverse biological matrices or in the context of complex drug regimens.

Response: We greatly appreciate the reviewer's feedback regarding the lack of discussion on the **real-world clinical applicability** of the GNPS Drug Library. We recognize the importance of clarifying this point and have now emphasized in the revised manuscript that the GNPS Drug Library is not intended for immediate clinical decision-making based on untargeted metabolomics data. Rather, it is designed as a research tool to support empirical drug exposure readouts and generate hypotheses that may ultimately inform future clinical studies. Several revisions were made to clarify these points.

Line 486-492: "We anticipate that the GNPS Drug Library will play a key role in clinical research studies by providing empirical drug exposure profiles. This capacity can enhance sample stratification and hypothesis generation, particularly for individuals with the same disease but differing medication regimens, as demonstrated in the above analysis on ARV therapy. While the GNPS Drug Library is not intended for immediate clinical decision-making, this tool lays the groundwork for future integration

of exposure data into precision medicine frameworks, such as studies of drug adherence, microbiome-drug interactions, and treatment heterogeneity.”

Line 525-533: “Finally, it is important to note that we do not anticipate the GNPS Drug Library to play an immediate role in clinical decision-making based on untargeted metabolomics profiles. However, the concepts introduced in this study will advance the use of untargeted metabolomics for personalized exposure assessment, pharmacokinetic profiling, and linking of exposures to biological responses, serving as foundational steps toward the broader goal of precision medicine, particularly by enabling exposure-informed analyses of clinical phenotypes and cohort substructure. We envision the GNPS Drug Library as a valuable resource supporting a wide range of scientific fields, ultimately driving new insights into both the ecological and health-related impacts of drug exposures.”

We revise the sentence in the abstract, “*The GNPS Drug Library holds potential for broader applications in drug discovery and precision medicine*” to improve clarity.

Line 84-88: “Overall, GNPS Drug Library provides a scalable resource for empirical drug screening in clinical, nutritional, environmental, and other research disciplines, facilitating insights into the ecological and health consequences of drug exposures. While not intended for immediate clinical decision-making, it supports data-driven exploration of drug exposures where traditional records are limited or unreliable.”

To demonstrate the utility of the GNPS Drug Library **across diverse biological matrices**, we first improved our analysis of the pharmacokinetic datasets by comparing the drug read-out with and without the drug metabolites/analogs in multiple biological matrices. We then provided two new analyses on drug detections, using the GNPS Drug Library, in the context of wastewater-based epidemiology and food screening, to demonstrate the broad implications of this resource across matrices and disciplines.

Line 317-339: “We analyzed two pharmacokinetic datasets where healthy individuals received specific probe drugs followed by time-series sampling. In the first study, 10 participants received a single oral dose of diphenhydramine. The drug was not detected in plasma and skin samples before administration, but was detected in all individuals post-administration over the course of 24 hours (Figure S4a,b). In plasma, detection frequencies peaked at 1-2 hours (Figure S4a,b), aligning with the reported time to maximum concentration (~2 hours) for diphenhydramine. In skin, peak detection occurred at 10-12 hours (Figure S4a,b), reflecting the delayed deposition to skin compared to plasma for orally administered drugs. In the second study, 14 participants received a cocktail of oral probe drugs, namely caffeine, midazolam, and omeprazole. The parent drugs were detected in plasma from 100% (caffeine), 46% (midazolam), and 100% (omeprazole) of participants within 8 hours post-administration, but were not detected in fecal samples (Figure S4c,e). Inclusion of drug metabolites and propagated analogs improved detection rates: midazolam detection rose to 69% in plasma and 7.1% in feces. Similarly, for omeprazole, fecal detection

increased from 0% to 21.4% when considering its metabolites and analogs (Figure S4d,f). Notably, at the 8-hour time point, 61.5% of participants exhibited detectable omeprazole only through its metabolites or analogs in plasma - highlighting the value of including derivative forms in drug exposure assessments. Together, these results underscore that drug detection is both biofluid- and time-dependent, and that metabolites and propagated analogs enhance detection sensitivity across sample types and time points. More broadly, the results emphasize the need to establish drug exposures empirically in the context of relevant biofluids – for example in feces when investigating microbiome-drug interactions - as clinical records do not account for drug distributions and rarely consider the time between drug intake and sample collection.

Revised Figure S4. Time-series drug detection in diverse biological matrices from healthy individuals receiving specific drugs in pharmacokinetic studies. **a-b**, Detection frequencies of diphenhydramine in plasma and skin samples from 10 individuals receiving a single dose of oral diphenhydramine (50 mg), based on parent compound only (**a**) or parent compound, metabolites, and analogs (**b**). The line chart demonstrates detections before (0 hour) and 24 hours (0.5, 1, 1.5, 2, 4, 6, 8, 10, 12, and 24 hours) after drug administration. **c-d**, Detection frequencies of the administered

drugs in plasma samples from 13 individuals receiving a cocktail of oral probe drugs, including caffeine (2 mg/kg), midazolam (0.075 mg/kg) and omeprazole (40 mg), based on parent compound only (c) or parent compound, metabolites, and analogs (d). Plasma samples were collected prior to (0 min) and 5 min, 30 min, 1, 2, 4, 5, 6, and 8 hours after administration of the drug cocktail. e-f, Detection frequencies of drugs in fecal samples from the 14 individuals receiving the drug cocktail, based on parent compound only (e) or parent compound, metabolites, and analogs (f). Caffeine was not detected in the fecal samples and therefore not included in panels e and f.

Line 106-108: “Outside of clinical settings, such as epidemiological monitoring of pharmaceuticals in wastewater, medical records are not available. These limitations highlight the need for direct data-driven approaches to screen drug exposures.”

Line 492-503: “In addition, the GNPS Drug Library is broadly applicable for drug screening across diverse research disciplines despite the focus on human biospecimen in this manuscript. For example, GNPS Drug Library can be applied for wastewater-based epidemiology to monitor population-level health trends. In wastewater samples collected from March to June 2020, we observed seasonality of drug usage - the abundances of a cough suppressant (dextromethorphan) and antibiotics decreased over time, while those of antihypertensives, antidepressants, HIV medications, and antiepileptics remain stable (Figure S9a). GNPS Drug Library can also be applied in the context of food monitoring. Using untargeted metabolomics files from ~3500 foods/beverages collected in the Global FoodOmics Project, we observed antibiotics (e.g., ampicillin, tetracycline) in fish, beef, and turkey, as well as antiparasitic drugs (e.g., spinosad, thiabendazole) that are also used as fungicides or insecticides in fruit and vegetables (Figure S9b-e).”

Revised Figure S9. Drug screening in wastewater and food samples using the GNPS Drug Library. a, Sample-to-sample peak areas of different drug classes in influent wastewater samples collected during March to June 2020 from three wastewater treatment plants (WWTPs) in Spain. Peak areas of multiple drugs in the same pharmacologic class were summed and standardized to the maximum value observed across all samples for this drug class. A non-parametric Kruskal-Wallis test followed by pairwise Wilcoxon test and Benjamini-Hochberg correction for multiple comparisons were performed. P-values < 0.05 were noted in the figure. Boxplots showcase the median value, first (lower) and third (upper) quartiles, and whiskers indicate the error range as 1.5 times the interquartile range. b-e, foodMASST search outputs showcasing detections of drugs in food products. Pie charts display the proportion of MS/MS matches found in the deposited reference database. Blue indicates a match with a food sample, while yellow represents a non-match. GFOP, Global FoodOmics Project.

Finally, we fully agree with the reviewer that stratification **in the context of complex drug regimens** is a key advance toward personalized medicine. In our HIV cohort, we leveraged this capability to uncover treatment-related heterogeneity in microbiome-linked lipid signals—an example of how empirical drug profiling can elucidate mechanistic differences within clinically homogeneous populations. This is now further highlighted in the manuscript.

Line 486-489: “We anticipate that the GNPS Drug Library will play a key role in clinical research studies by providing empirical drug exposure profiles. This capacity can enhance sample stratification and hypothesis generation, particularly for individuals with the same disease but differing medication regimens, as demonstrated in the above analysis on ARV therapy.”

Insufficient Validation - Although the library offers insights into drug exposure trends by demographics and disease states, these findings rely heavily on assumptions that the detected molecules accurately reflect drug exposure. A rigorous validation process is noticeably absent, raising questions about the reliability of these results. Moreover, the inclusion of only 10% of drugs with metabolite reference spectra further limits the library’s utility, particularly for assessing drug metabolism comprehensively.

Response: We fully agree with the reviewer that **validation** of drug exposures detected with GNPS Drug Library, and annotations from untargeted metabolomics in general, is critical. Since the drug exposure trends by demographics and disease states were derived by analyzing public metabolomics data, we were not able to further validate these results with analytical standards. However, we note that analysis across such diverse demographics and disease states is made possible only by utilizing public metabolomics data. We now emphasized this limitation in the revised manuscript. We also note that important medications detected in the HNRC cohort are validated with analytical standards (Figure S10).

To reiterate from the clinical perspective, the GNPS Drug Library is not presented as a tool for patient-level treatment decisions. Rather, it serves as a scalable, empirical resource to enhance research on treatment patterns, adherence, and off-target pharmacologic effects—areas that are increasingly critical as we move toward data-driven cohort stratification in precision health research.

Line 369-373: “We note that the re-analysis of public metabolomics datasets is based on MS/MS matches, thus corresponding to level 2/3 confidence according to the 2007 Metabolomics Standards Initiative. Due to practical challenges to access sample extracts for >20 studies discussed in the above analyses, we were not able to further validate the annotations with analytical standards.”

Line 509-513: “It is important to understand that the use of the GNPS Drug Library holds certain limitations. The current library only supports MS/MS-based annotations to level 2/3 according to the 2007 Metabolomics Standards Initiative. This generally means that spectra of drug isomers may be annotated as the drug. Key drugs of

interest based on the matches obtained from this resource should be checked for retention time and MS/MS matching with analytical standards.”

For inclusion of only 10% of drugs with metabolite reference spectra, we want to first clarify that the metabolite mining effort focused only on drugs without endogenous and food sources. Combining the known metabolites and propagated drug analogs, metabolite reference spectra are available for 18% of exogenous drugs in our original version of this resource.

In addition, to further **expand the coverage of the drug analog library**, we leveraged recent development of the fastMASST search tool and **fully updated the GNPS Drug Analog Library**. Our original drug analog search, conducted in June 2023, only mined public datasets on the GNPS/MassIVE repository before the date (~1,700 datasets). For this updated analog library in March 2025, provided in the revised manuscript, we mined not only the additional datasets uploaded to GNPS/MassIVE from June 2023 to March 2025, but also all the metabolomics datasets on the MetaboLights and Metabolomics Workbench repositories, other two of the largest public repositories for metabolomics data that were now searchable using fastMASST (~3,500 datasets mined in total). The updated drug analog library contains 12,455 clustered MS/MS spectra (rather than repeated MS/MS spectra for the same molecule) for 1,277 drugs. Now, combining with known metabolites, metabolite/analog reference spectra are available for 37% of exogenous drugs.

Although 37% is arguably not an amazingly high coverage of drug metabolites, our work represents the first resource of its kind to discover drug metabolites from public metabolomics studies at a large scale. Without this approach, drug metabolites need to be synthesized or generated *in vitro* before reference spectra can become available, which is low-throughput and the reason why there are so few reference spectra for drug metabolites publicly available. We also want to emphasize that not all drugs are metabolized. For example, Ross et al. (*J. Am. Soc. Mass Spectrom.* 2022, 33, 1061–1072) incubated 2,000 drugs with human liver microsomes and S9 fraction, and identified 572 metabolites. The MS/MS spectra of drug metabolites from Ross et al. are included in the GNPS Drug Library.

Finally, we clarified that all the analysis in the manuscript has been updated with the most recent version of the drug analog library.

Technical Limitations - A critical technical shortcoming is the failure to address in-source fragmentation (ISF), a known and pervasive issue in small molecule mass spectrometry. ISF artifacts can masquerade as genuine metabolites, leading to false positives. GNPS has historically struggled with this issue, as evidenced by its cataloging of over a billion spectra (Cell, 2024) the majority of which is now recognized as artifactual. Without explicit

methodologies to mitigate ISF, the GNPS Drug Library risks introducing significant inaccuracies, undermining its ability to support empirical drug exposure assessments.

Response: We fully agree with the reviewer that we did not address the analytical artifacts that result in multiple ion forms per molecule in sufficient detail in the original manuscript. We now provided a detailed evaluation on the proportion of drug analogs possibly derived from analytical artifacts in **Figure S1**. We like to point out, however, that in real data these also exist and by including them in the library, they help rather than prevent drug detection. Omitting them would be counter to what the resource is aiming to achieve. Based on peak shape correlation and fragment matching strategies (see details in Method section), we determined that ~11% of the propagated drug analogs are in-source fragments of the parent drugs.

Figure 1d: “Total unique analogs: 12,455; In-source fragments: 1,428; Isotopes: 523; Adducts: 2,179.”

Line 266-272: “Based on peak shape correlation and fragment matching analysis, 33% of the propagated drug analogs could be other ion forms of the parent drug, including isotopes (5%), adducts (17%), or in-source fragments (11%), while the rest 67% are likely drug metabolites or structural analogs (Figure 1d, S1). Although the propagated analogs include non-biological ion forms such as adducts and in-source fragments, we retain them in the library to enhance detection sensitivity. Their presence still signals drug exposure even if they do not represent metabolic derivatives.”

Revised Figure S1. Estimated proportions of isotopes, adducts, in-source fragments, and drug metabolites or structural analogs in the GNPS propagated drug analog library. Drug analogs were first assigned as isotopes or adducts (sodium, calcium, and potassium adducts) based on mass offsets with parent drugs. Then, for analogs that occurred in the same public metabolomics data with the parent drugs, peak shape correlations were employed to assign the drug analogs as instrumental artifacts (Pearson correlation $R^2 > 0.9$) or drug derivatives (Pearson correlation $R^2 < 0.9$). Analogues were classified based on the maximum R^2 observed across multiple

data files. For analogs that did not co-occur with the parent drugs, they were classified based on MS/MS fragment matching. Specifically, for analogs with precursor m/z higher than the parent drugs, they were classified as adducts if any of their MS/MS fragment masses match the drug precursor m/z. For analogs with precursor m/z lower than the parent drug, they were classified as in-source fragments if their precursor m/z matches any of the MS/MS fragments of the drugs. No intensity filtering of the MS/MS spectra were used in this step.

Misleading Emphasis on Spectral Data Quantity - The authors emphasize the library's scale, citing 99,000 spectra for 4,000 molecules. However, this focus on the number of spectra is misleading. Modern mass spectrometers can generate millions of spectra from a single molecule in days. The more relevant metric is the number of unique, high-quality molecular identifications, which is less impressive and highlights the library's current limitations.

Response: We fully agree with the reviewer that emphasis on the number of spectra is unnecessary. We reduced the emphasis on spectra numbers throughout the manuscript, especially in Figure 1a. We also want to note that this collection of GNPS Drug Library represents all publicly accessible drug MS/MS spectra. We especially included spectra from the newly generated MSⁿLib, which was developed with analytical standards of ~20,000 compounds, together with ~50,000 unique structures covered by GNPS and MassBank. Notably, these public, community-sourced reference spectra were collected on diverse instruments and collision energies from labs across the world. Such diversity greatly enhances the utility for this library to the research community, compared with reference libraries generated in a single lab on a single instrument. In addition, we want to point out that the drug analog library contains 12,455 unique MS/MS spectra, post spectral merging. Therefore, they do not represent spectra generated from the same molecules over and over again.

Line 165-167: "This represents all publicly accessible drug MS/MS spectra, covering 4,723 unique drugs (represented by 99,122 MS/MS spectra) analyzed with diverse instrumentation and collision energies."

Line 181-182: "In total, we collected metabolite spectra for 470 drugs in the GNPS Drug Library (represented by 2,080 MS/MS spectra; **Figure 1a**)"

Line 259-260: "After the filtering steps, propagated analogs of 1,277 drugs (12,455 clustered non-duplicated MS/MS spectra) were collected in the final drug analog library."

Revised Figure 1. The GNPS Drug Library and connected pharmacologic metadata.

a, The GNPS Drug Library comprises four key resources: Drug MS/MS reference spectra, drug metabolite MS/MS reference spectra, propagated drug analogs derived

from public metabolomics datasets, and pharmacologic metadata connected to each reference spectrum.

Overall - Despite its potential, the GNPS Drug Library is far from ready for clinical application. The absence of rigorous validation, especially against clinical standards, combined with the lack of attention to ISF and other technical challenges, undermines its reliability. At this stage, relying on this tool for high-stakes decisions in precision medicine would be risky at best.

Response: We thank the reviewer for the insightful comments and fully acknowledge the need for greater clarity. As detailed in our prior responses, we recognize that our original language may have unintentionally conveyed the impression that the GNPS Drug Library is intended to support clinical decisions for individual patients. We have worked to strike a clear boundary between current applications and aspirational goals. The GNPS Drug Library is not a diagnostic assay; it is a tool for identifying patterns of real-world drug exposure with sufficient fidelity to guide hypothesis generation in clinical research studies, particularly in the context of neuroinflammation, polypharmacy, and pharmacomicrobiomics.

We have adjusted the overall text to eliminate the discussion emphasis of clinical application in precision medicine - except in the limitations section (see below) which was written by the clinicians on the paper as they see this resource as a key stepping stone to enable medication exposure readout in their clinical studies in the future. It now reads

“While it is not intended for immediate clinical decision-making, this tool lays the groundwork for future integration of exposure data into precision medicine frameworks, such as studies of drug adherence, microbiome-drug interactions, and treatment heterogeneity.”

“The concepts introduced in this study will advance the use of untargeted metabolomics for personalized exposure assessment, pharmacokinetic profiling, and linking of exposures to biological responses, serving as foundational steps toward the broader goal of precision medicine, particularly by enabling exposure-informed analyses of clinical phenotypes and cohort substructure.”

In addition, we validated key medications with analytical standards, and we have updated the drug analog library (which will be continuously updated over the years, especially as new drug reference MS/MS become available) to increase the analog/metabolite coverage to 37% of the drugs. We further provided a detailed evaluation on the possible in-source fragments for the updated propagated drug analog library.

**Point-by-point response to reviewer comments for:
A resource to empirically establish drug exposure records directly from untargeted metabolomics data**

EDITORIAL DECISIONS:

Thank you for submitting your manuscript "A resource to empirically establish drug exposure records directly from untargeted metabolomics data" to Nature Communications. I am delighted to say that we are happy, in principle, to publish it under an open access license.

First, we ask you to revise your paper to address our editorial requests (in the attached Author Checklist) and any remaining comments from reviewers (included at the end of this email, if applicable).

Failure to comply with our editorial requests will result in further revisions and delays in accepting your manuscript. Please also see the *Nature Communications* formatting instructions for further information.

We look forward to seeing your revised manuscript. Please let us know if the process may take longer than two weeks.

Response: We really appreciate the efforts from the editorial team and are very grateful for the chance to publish in *Nature Communications*. We have revised the manuscript according to the editorial requests and addressed the reviewer concerns to improve clarity. Please see detailed response to reviewer comments below.

REVIEWERS' COMMENTS

Reviewer #1 (Remarks to the Author):

The authors did a great job to address my previous comments and now the paper have been largely improved. I have no further comments.

Response: We thank the reviewer for supporting our work and the valuable inputs during the review process!

Reviewer #2 (Remarks to the Author):

"A resource to empirically establish drug exposure records directly from untargeted metabolomics data"

This revised submission expands upon the earlier version, with additions such as an enlarged GNPS Drug Analog Library, an in-source fragment analysis, broader scope beyond clinical applications, and incorporation of plasma datasets alongside fecal metabolomics. While these changes address some prior concerns, they do not resolve several fundamental issues that limit the overall utility of the approach.

Response: We are grateful that the reviewer noted our efforts to expand the GNPS Drug Analog Library, refine the in-source fragment interpretation, broaden the study context, and add plasma metabolomics datasets. We have addressed the remaining points by the reviewer.

Validation of analog/metabolite identifications

The co-occurrence analysis (63% overall) is an indirect and insufficient validation strategy. Without systematic confirmation using authentic standards or orthogonal methods, there remains substantial risk of major misannotations, mis-assignment of in-source fragments, or detection of unrelated co-eluting compounds. The explanation that absence of the parent drug may reflect complete metabolism is speculative and unsubstantiated without direct evidence.

Response: We appreciate the reviewer's concern regarding validation. The GNPS Drug Analog Library is designed as a first-pass discovery resource to enable broad screening and hypothesis generation, rather than as a definitive source of validated metabolite identifications. As is standard in untargeted metabolomics, subsequent confirmation with authentic standards or orthogonal methods remains essential before translation into clinical or mechanistic studies. We explicitly demonstrated this workflow in the HIV cohort analysis, where several drug metabolites were validated by comparing both retention time and MS/MS

spectra against microbial culture products. These analogs displayed distinct retention times relative to parent drugs and were directly observed in microbial culture experiments only when the drug is added, supporting their biological origin rather than being artifacts of in-source fragmentation.

To aid users in distinguishing candidate metabolites from potential artifacts, we benchmarked the analog library and estimated that ~33% of the entries can be attributed to in-source fragments, adducts, or isotopes. We have now included this information in the library metadata, specifying the potential source category (in-source fragment, adduct, structural analog, etc.) for each entry. This update is described in the revised manuscript and will help users interpret annotations in the appropriate discovery context.

Line 283-285: “The source (as in-source fragments, adducts, or structural analogs) and the availability of ModiFinder prediction for each drug analog were provided in the GNPS Drug Library metadata (see Data Availability section).”

Example screenshot of the updated Analog Library metadata:

A	B	D	G	H	I	J	K	L	M
analog_libid	name_analog	delta_mass	final_class	modifier	name_conf	pharmacol	therapeutic	therapeutic	mechanism
CCMSLIB00013654469	related spectra of 'darunavir (delta mass:-2.016)'	-2.02	Drug derivatives	no	darunavir	HIV protease	infectious dis	human immu	HIV protease
CCMSLIB00013654470	related spectra of 'darunavir (delta mass:15.995)'	15.99	Possible Adducts	no	darunavir	HIV protease	infectious dis	human immu	HIV protease
CCMSLIB00013654471	related spectra of 'darunavir (delta mass:29.974)'	29.97	Drug derivatives	no	darunavir	HIV protease	infectious dis	human immu	HIV protease
CCMSLIB00013654472	related spectra of 'darunavir (delta mass:162.053)'	162.05	Drug derivatives	no	darunavir	HIV protease	infectious dis	human immu	HIV protease
CCMSLIB00013654473	related spectra of 'darunavir ethanolate (prezista) (c -2.02)'	-2.02	Drug derivatives	yes	darunavir	HIV protease	infectious dis	human immu	HIV protease
CCMSLIB00013654474	related spectra of 'darunavir ethanolate (prezista) (c 15.99)'	15.99	Possible Adducts	yes	darunavir	HIV protease	infectious dis	human immu	HIV protease
CCMSLIB00013654475	related spectra of 'darunavir ethanolate (prezista) (c 176.03)'	176.03	Possible Adducts	no	darunavir	HIV protease	infectious dis	human immu	HIV protease
CCMSLIB00013654708	related spectra of 'massbank:au217909 darunavir '	15.99	Drug derivatives	yes	darunavir	HIV protease	infectious dis	human immu	HIV protease
CCMSLIB00013654853	related spectra of 'massbank:eq301602 darunavir '	-43.99	Possible ISF	no	darunavir	HIV protease	infectious dis	human immu	HIV protease
CCMSLIB00013654854	related spectra of 'massbank:eq301602 darunavir '	-2.02	Drug derivatives	yes	darunavir	HIV protease	infectious dis	human immu	HIV protease
CCMSLIB00013654855	related spectra of 'massbank:eq301602 darunavir '	15.99	Possible Adducts	yes	darunavir	HIV protease	infectious dis	human immu	HIV protease
CCMSLIB00013654856	related spectra of 'massbank:eq301602 darunavir '	29.97	Drug derivatives	yes	darunavir	HIV protease	infectious dis	human immu	HIV protease
CCMSLIB00013654857	related spectra of 'massbank:eq301602 darunavir '	162.05	Drug derivatives	no	darunavir	HIV protease	infectious dis	human immu	HIV protease

Applied and clinical relevance

While the authors now state the resource is “not for immediate clinical decision-making,” the examples remain largely descriptive, with no compelling demonstration that the resource changes outcomes compared to existing approaches. The manuscript would benefit from a concrete, integrated example—such as showing how library-based exposure detection alters patient stratification, epidemiologic interpretation, or experimental conclusions. As it stands, the environmental and food monitoring examples feel tangential rather than woven into a coherent scientific narrative.

Response: We fully agree with the reviewer that an integrated sample, showing library-based exposure detection aids patient stratification, is the key of this resource. This is exactly what we highlight with the HIV-medication stratification example. As explained in Line 482-488, “For example, metadata for the HNRC cohort on current ARV usage, which is based on self-reports, documented drug usage as “ARV-naïve” (never received ARV), “no ARV” (no current ARV use), “non-HAART” (currently using less than three ARVs), and “HAART” (currently using three or more ARVs). Based on these classifications, no significant differences were

observed for the N-acyl lipids detected in these samples (Figure S8b). Without the empirical drug readout, enabled by the GNPS Drug Library, the effects of drugs on microbial N-acyl lipid levels were overlooked.”

In addition, “medication records” might simply not be available in scenarios such as environmental samples or food monitoring. Therefore, they are essential examples to demonstrate the broader usage of GNPS Drug Library.

We revised the abstract and text on the food and environmental examples to clarify the implication of the library-based approach.

Line 83-85: “We demonstrate its application by stratifying participants in a human immunodeficiency virus (HIV) cohort based on detected drug exposures. We uncover drug-associated alterations in microbiota-derived N-acyl lipids that were not captured when stratifying by self-reported medication use.”

Line 513-515: “Despite the focus on human biospecimen in this manuscript, the GNPS Drug Library is broadly applicable for drug screening across diverse research disciplines, such as food and environmental monitoring where medical records are not available.”

Sampling and detection gaps

The additional plasma analyses (Figure S5a) are retrospective and limited in scope, leaving unresolved the concern that fecal metabolomics misses certain exposures. Only a prospective, paired fecal/plasma study within the same cohort could convincingly address this limitation.

Response: We agree that a prospectively designed, paired fecal/plasma study would provide the most comprehensive evaluation of biospecimen-specific exposure detection. Conducting such a study, however, would require initiating a full clinical trial, which is beyond the scope of this resource introduction. In the present work, we leveraged an existing Cooperstown cocktail study, which involved administration of multiple drugs with and without antibiotics, longitudinal collection of both fecal and plasma samples, and systematic data acquisition. While our analyses of these data were retrospective, the study design nevertheless captures the same fundamental contrasts that a prospective trial would address - namely, how exposures are differentially detectable across biospecimens using this resource.

Our goal in this manuscript is to introduce and benchmark the GNPS Drug Library as a broadly applicable discovery resource, rather than to comprehensively resolve biospecimen-specific coverage. We believe the Cooperstown analysis effectively illustrates these principles while providing a foundation for future, purpose-built prospective studies.

Quantification and dosage estimation

The authors acknowledge that untargeted metabolomics cannot provide dosage estimation, yet this absence substantially restricts the real-world applicability of the approach. No example is offered to show how this gap might be mitigated in practice, leaving a critical part of exposure interpretation unresolved.

Response: We agree with the reviewer that untargeted metabolomics does not provide absolute dosage information. However, it does yield within-study, individualized, relative quantitative drug exposure profiles, which are often sufficient to stratify participants and uncover biologically meaningful differences. In many research settings, distinguishing higher versus lower exposure levels is more relevant than knowing exact dosage.

As shown in our HIV cohort analysis, these relative exposure profiles enabled participant stratification and revealed drug-associated metabolic alterations that were not captured by self-reported medication use. We have clarified this point in the revised manuscript:

Line 506-510: “We anticipate that the GNPS Drug Library will play a key role in clinical research studies by providing individualized, within-study relative quantitative drug exposure profiles. This capacity can enhance sample stratification and hypothesis generation, particularly for individuals with the same disease but differing medication regimens or pharmacokinetics, as demonstrated in the above analysis on ARV therapy.”

Library expansion and quality control

The growth from ~3,200 to ~12,400 analog spectra is notable, but the manuscript lacks transparency on quality control: no clear confidence metrics, false discovery rates, or proportions of high-confidence vs. putative entries are provided. The continued inclusion of spectral counting, previously noted as largely meaningless, further detracts from the rigor. A breakdown of analog source types, validation levels, and associated confidence would be essential for users to gauge reliability.

Response: In the previous round, in response to the reviewer comments, we had removed the spectral counts of the library from Figure 1 and most of the text. We however, retained them only in one instance to illustrate the current scale of the library. While we agree the number of unique compounds is the most important metric compared to the counts of spectra, we think they still reflect the scale of the library that is currently available (although will be growing significantly in the future), because they are publicly sourced and generated on different instruments, different collision energies and labs, rather than repetitive entries. This is important for the broadest ability of the library to detect drugs from data obtained from many different platforms.

But most importantly, we emphasize that the GNPS Drug Library is intended as a discovery resource for first-pass annotation. As demonstrated in the HIV study, library matches should always be followed by verification with authentic standards when possible. Users can apply any scoring thresholds, though our typical criteria - cosine score ≥ 0.7 and a minimum of six matching ions - yield a false discovery rate of 1% (Scheubert et al., *Nat. Commun.* **2017**, PMID: 29133785). This is consistent with established practice in untargeted metabolomics MS/MS matching. When retention time and MS/MS are further confirmed against authentic standards, annotations can be advanced to level 1 according to the Metabolomics Standards Initiative guidelines (Sumner et al., *Metabolomics* **2014**, PMID: 24039616).

To provide clarity to the reviewer and the readers, we have now explicitly adjusted the following statement.

Line 531-536: “As with all MS/MS library matches in untargeted metabolomics, annotations from the GNPS Drug Library should be considered structural hypotheses derived from MS/MS spectral matches. To obtain definitive identifications, key drugs and metabolites of interest must be validated by authentic analytical standards through retention time and MS/MS comparison to achieve Level 1 identification,⁵⁹ or by orthogonal strategies such as in vitro culturing when authentic standards are not available, particularly for drug metabolites.”

Overall, while the authors have made additions, the work still suffers from fundamental weaknesses: indirect and incomplete validation, absence of quantification, limited evidence for practical impact, and insufficient transparency in library quality. These limitations (many acknowledged by the authors themselves) restrict the current utility of the approach for clinical, epidemiologic, or environmental applications.

Response: We thank the reviewer for their careful assessment. As detailed in our point-by-point responses above, we have addressed the concerns and improved clarity. We emphasize that the GNPS Drug Library is introduced here as a discovery resource - a foundation to enable exposure detection and hypothesis generation - rather than as a fully validated clinical or regulatory tool. Within this scope, we believe the revisions now demonstrate the rigor, transparency, and practical utility appropriate for a resource paper, while also outlining clear directions for future confirmatory and translational studies.